# Collaborative Cognitive Diagnosis with Disentangled Representation Learning for Learner Modeling

Weibo Gao[1]   Qi Liu[1,2]*   Linan Yue[1]   Fangzhou Yao[1]   Hao Wang[1]
Yin Gu[1]   Zheng Zhang[1]

[1] State Key Laboratory of Cognitive Intelligence, University of Science and Technology of China
[2] Institute of Artificial Intelligence, Hefei Comprehensive National Science Center
weibogao@mail.ustc.edu.cn; qiliuql@ustc.edu.cn; {lnyue, fangzhouyao}@mail.ustc.edu.cn;
wanghao3@ustc.edu.cn; {gy128, zhangzheng}@mail.ustc.edu.cn

## Abstract

Learners sharing similar implicit cognitive states often display comparable observable problem-solving performances. Leveraging collaborative connections among such similar learners proves valuable in comprehending human learning. Motivated by the success of collaborative modeling in various domains, such as recommender systems, we aim to investigate how collaborative signals among learners contribute to the diagnosis of human cognitive states (i.e., knowledge proficiency) in the context of intelligent education. The primary challenges lie in identifying implicit collaborative connections and disentangling the entangled cognitive factors of learners for improved explainability and controllability in learner Cognitive Diagnosis (CD). However, there has been no work on CD capable of simultaneously modeling collaborative and disentangled cognitive states. To address this gap, we present Coral, a Collaborative cognitive diagnosis model with disentangled representation learning. Specifically, Coral first introduces a disentangled state encoder to achieve the initial disentanglement of learners' states. Subsequently, a meticulously designed collaborative representation learning procedure captures collaborative signals. It dynamically constructs a collaborative graph of learners by iteratively searching for optimal neighbors in a context-aware manner. Using the constructed graph, collaborative information is extracted through node representation learning. Finally, a decoding process aligns the initial cognitive states and collaborative states, achieving co-disentanglement with practice performance reconstructions. Extensive experiments demonstrate the superior performance of Coral, showcasing significant improvements over state-of-the-art methods across several real-world datasets. Our code is available at `https://github.com/bigdata-ustc/Coral`.

## 1 Introduction

It is a common notion that individuals with similar implicit states frequently exhibit similar explicit behaviors. Therefore, establishing interconnections among similar users is crucial for understanding human behaviors. For instance, social connections play a pivotal role in understanding current consumer preferences and predicting future behaviors [42]. Similarly, in the

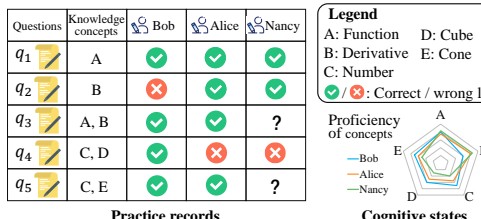

Figure 1: An example of human learning, where learners individually select questions to practice. Each question tests at least one knowledge concept.

---

*Indicates Corresponding Author.

context of intelligent education, a better modeling of like-minded learners with similar learning experiences, is essential for understanding the human learning process [31], analyzing their knowledge proficiency and facilitating personalized tutoring tailored to individual needs [55].

As illustrated in Figure 1, we can infer Nancy is likely to answer the *Cone*-related question $q_5$ correctly according to the correct practice responses of Bob and Alice, who share similar learning behaviors with Nancy. The underlying psychological assumption is that learners with similar experiences generally possess similar cognitive states — how well the learner masters each knowledge concept, influencing their subsequent responses. To gain a deeper understanding of the human learning process, it is crucial to explicitly diagnose unobservable cognitive states. Existing Cognitive Diagnosis (CD) methods seek to enhance diagnostic accuracy by fully utilizing the inner-learner information (i.e., individual attributions [41] and explicit practice records) and question-side features (e.g., difficulty [16], textual content [26], and educational relations [12, 14]). However, the issue of how similar (a.k.a. collaborative) connections among inter-learners with similar states facilitate understanding of learners' knowledge proficiency remains largely unexplored.

In this study, to efficiently harness collaborative information among similar learners and thereby more accurately diagnose the cognitive states of each individual, we advocate for the incorporation of inter-learner connections into the CD process. However, designing a collaborative CD model in educational scenarios presents two distinct challenges due to the complexity of human learning.

- First, acquiring explicit collaborative connections among learners proves to be a formidable challenge. On the one hand, unlike many well-defined social scenarios (e.g., *Twitter.com*), where user preference similarities are manifested through explicit social actions such as following and liking, the directly available social behaviors among learners in learning environments (e.g., *LeetCode.com*) cannot be used for diagnosis modeling since these social attributes cannot reflect true cognitive-oriented connections. On the other hand, some related studies [28, 13] attempt to design different similarity functions based on practice data to compute cognitive similarities among learners. However, these approaches pose a significant challenge of manually selecting appropriate metrics and corresponding thresholds, introducing additional inductive biases. Although various methods for constructing user relationships have been proposed in other domains [22, 9], these approaches do not consider the domain-specific attributes of students in learning scenarios and cannot be directly applied in educational contexts.

- Second, an ideal collaborative diagnosis procedure requires disentangling and uncovering the mixed explanatory latent factors hidden in the observed learning behaviors. The basic motivation is that learners demonstrate complex and diverse patterns driven by entangled states across both inner- and inter-learner perspectives. For instance, from an inner perspective, Nancy may not master *Cone* since she does not practice *Cone*-related questions. However, based on inter-learner data, one can infer a high probability that she has mastered *Cone*. Most prior attempts can not fulfill this requirement since they learn representations in an entangled way. Although recent models [8, 50] achieve a dimension-level disentanglement of cognitive states, they lack consideration of modeling the influence of collaborative connections, ignoring the complex relations between inner- and inter-learner connections of different individuals. Thereby, it needs to find a suitable way to achieve the co-disentanglement from both the inner- and inter-learner views for cognitive representations with higher interpretability and controllability.

To tackle the above challenges, we propose Coral, a Collaborative cognitive diagnosis model with disentangled representation learning, to reveal learner cognitive states while simultaneously modeling both inner- and inter-learner learning information. Specifically, our approach begins with the disentangled cognitive representation encoding to establish initial disentangled learner states through reconstructing their practice performance from the inner-learner perspective. Next, our focus shifts to effectively learning collaborative cognitive representations from the inter-learner perspective. The most significant point is to find the implicit collaborative relations between learners. To address this challenge, we present a context-aware collaborative graph learning mechanism that automatically explores all $K$-optimal neighbors for each learner given their basic cognitive states to facilitate the explicit modeling of collaborative connections among learners. Based on the constructed graph, collaborative information can be effectively fused into disentangled learner cognitive states through learning collaborative node representations. Finally, a decoding and reconstruction process is conducted to merge initial states and collaborative states so as to achieve co-disentanglement from both the inner- and inter-learner perspectives. Extensive experiments demonstrate the superior performance of Coral, showing significant improvements over SOTA methods across several datasets.

## 2   Related Work

**Cognitive Diagnosis** As a fundamental task, cognitive diagnosis (CD) has been well-researched for decades in educational psychology [25, 4, 53]. It aims to profile the implicit cognitive states (i.e., the proficiency of specific knowledge concepts) of learners by exploiting observed practice records (e.g., correct or wrong). Existing research on CD assumes that learners' knowledge proficiency is proportional to their practice performance and thus can be diagnosed through predicting their practice performance [12]. Since the diagnostic results can be applied to many intelligent applications, such as exercise recommendation [18] and learning path suggestions [55], many CD models have been proposed in recent years. The early works from psychology like IRT [16] and MIRT [1] focus on modeling learners' answering process by predicting the probability of a learner answering a question correctly, which utilizes latent factors as the learner's ability. These methods lack interpretability, i.e., they are inability to output explicit multidimensional diagnostic results on each knowledge concept. To achieve better interpretability, later diagnostic models focus on incorporating knowledge concepts of questions to diagnose learners' proficiency on all knowledge concepts [38, 45, 46, 32]. Representative NCDM [40] adopts neural networks to model non-linear interactions instead of handcrafted interaction functions in previous works (e.g., IRT, and MIRT). In summary, existing CD studies enhance diagnostic accuracy by fully utilizing the inner-learner information (i.e., individual attributions and explicit practice records) [41, 50]and question-side features (e.g., difficulty [16, 38], textual content [26], and educational relations [12, 14, 8]). However, to the best of our knowledge, the problem of collaborative diagnostic modeling remains largely unexplored.

**Collaborative modeling in Education** Collaborative connections among learners in the education context commonly refer to learners with similar explicit practice behaviors, testing scores and implicit knowledge proficiency [28, 54, 44]. However, due to the complexity and implicitness of the human learning process, these relations are commonly not explicitly and directly available. Existing studies [28, 13] in AI Education have attempted to design different similarity functions based on practice data to compute cognitive similarities among learners. However, these methods pose a significant challenge of manually selecting appropriate metrics and corresponding thresholds, introducing additional inductive biases.

**Disentangled Representation Learning** Disentangled Representation Learning (DRL)[3], which aims to produce robust, controllable, and explainable representations, has become one of the core problems in machine learning. Typical methods include variational method [20], weakly supervised models [21], as well as the recent combination with the diffusion model [6]. DRL has a wide range of applications in user modeling to disentangle attributes. For example, recommendation with several aspects of users' interests [24, 33], fair user representation to disentangle sensitive attributes [10]. In education, DCD [8] attempts to disentangle learners' cognitive representations via variational framework, which motivates us to conduct a further study on collaborative CD setups.

## 3   Coral

We first introduce the problem setup, followed by details on three core components of Coral: i) Disentangled Cognitive Representation Encoding, ii) Collaborative Representation Learning and iii) Decoding and Reconstruction. Figure 2 shows the framework. The algorithm is listed in Algorithm 1.

### 3.1   Problem Setup

Our setup considers the human learning dataset $D$ including the practice records between $M$ learners and $N$ questions. The practice records of each learner $u$ is denoted by $\mathbf{x}_u = \{x_{u,i}\}$, where $x_{u,i}$ equals 1 or 0, representing that learner $u$ answered question $i$ correctly or not, respectively. Each question is related to at least one knowledge concept. The association relations between $N$ questions and $C$ knowledge concepts is represented by $\mathbf{C} = \{\mathbf{c}_i\}_{i=1}^N$, where $\mathbf{c}_i \in \mathbb{R}^C$ and $c_{i,c}$ equals 1 or 0 denoting that question $i$ is related to concept $c$ or not. The practice records are regarded as the explicit inner-learning information in our context.

Besides, we consider the collaborative connections among learners with similar cognitive states, which provide the inter-learner information. We define collaborative connections as a graph structure $G = (V, E)$ which contains a set of nodes (i.e., learners) $V$ and a set of edges $E$ where $(u, v) \in E$ or $(u, v) \in G$ indicates that the existence of a collaborative connection between learner $u$ and $v$ (i.e., $u$ and $v$ have similar latent cognitive states). Notably, the collaborative connections in educational

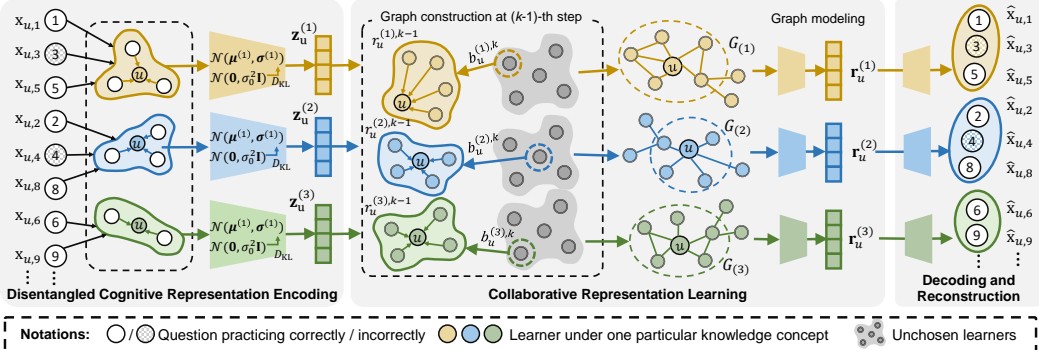

Figure 2: The overall framework of Coral.

scenarios are generally not explicitly or directly available, and it needs to design an adaptive strategy to automatically infer similar learners from observed learning data during the training process.

To achieve cognitive state disentanglement, we initially assign $C$ factorized representations to each learner, i.e., $\mathbf{z}_u = [\mathbf{z}_u^{(1)}; \mathbf{z}_u^{(2)}; \ldots; \mathbf{z}_u^{(C)}] \in \mathbb{R}^{d \times C}$ with Gaussian Mixture initialization since the Gaussian distribution has long been recognized as a proper statistic model for the cognitive states of learners in educational psychology [4]. The component $\mathbf{z}_u^{(c)}$ is expected to capture the learner $u$'s cognitive state over knowledge concept $c$. We denote $\Theta$ as the set of trainable parameters for the proposed model. Based on the above setups, the goal of Coral is to learn co-disentangled representations $\tilde{\mathbf{Z}} = \{\tilde{\mathbf{z}}_u\}_{u=1}^M$ for the $M$ learners from both the inner-learner practice perspective and inter-learner collaborative perspective.

## 3.2 Disentangled Cognitive Representation Encoding

The practice response $\mathbf{x}_u$ of each learner $u$ provides valuable inner-learner insights regarding his/her proficiency since learners' performance on each question is assumed to be proportional to their cognitive proficiency on question-related knowledge concepts [8]. Therefore, we implement an encoder for encoding the disentangled cognitive state $\mathbf{z}_u$ of each learner $u$ by reconstructing their practice responses. For a learner $u$, we assume that his/her practice performance on candidate questions can be generated from the following distribution:

$$p_\Theta(\mathbf{x}_u) = \mathbb{E}_{p(\mathbf{C})} \left[ \int p_\Theta(\mathbf{x}_u \mid \mathbf{z}_u, \mathbf{C}) \, p_\Theta(\mathbf{z}_u) d\mathbf{z}_u \right], \tag{1}$$

where $p(\mathbf{C}) = p_D(\mathbf{C})$ and $p_\Theta(\mathbf{x}_u \mid \mathbf{z}_u, \mathbf{C})$ is naturally a cognitive diagnosis procedure to predict practice performance. The key point of this task is to learn an optimal encoder $p_\Theta(\mathbf{z}_u)$ via practice records $\mathbf{x}_u$ to encode the cognitive state $\mathbf{z}_u$ of each learner $u$. To optimize $\Theta$, we introduce a variational distribution $q_\Theta(\mathbf{z}_u \mid \mathbf{x}_u)$ to approximate $p_\Theta(\mathbf{z}_u)$, following the VAE literature [3], through maximizing a lower bound of $\log p_\Theta(\mathbf{x}_u)$ based on the following property.

**Property 1.** $\max \log p_\Theta(\mathbf{x}_u)$ *is bounded as follows:*

$$\log p_\Theta(\mathbf{x}_u) \geq \mathbb{E}_{p(\mathbf{C}) q_\Theta(\mathbf{z}_u \mid \mathbf{X}_u)} \left[ \log p_\Theta(\mathbf{x}_u \mid \mathbf{z}_u) \right] - \mathbb{E}_{p(\mathbf{C})} \left[ D_{\mathrm{KL}}(q_\Theta(\mathbf{z}_u \mid \mathbf{x}_u) \| p_\Theta(\mathbf{z}_u)) \right]. \tag{2}$$

See the Appendix A for the proof.

In Property 1, the first term reconstructs the true practice performance $\mathbf{x}_u$ of learner $u$ and the variational encoder $q_\Theta(\mathbf{z}_u \mid \mathbf{x}_u)$ in the second term approximates the true encoder $p_\Theta(\mathbf{z}_u)$ by minimizing the KL divergence $D_{\mathrm{KL}}$. The variational distribution $q_\Theta(\mathbf{z}_u \mid \mathbf{x}_u)$ and the expectation $\mathbb{E}_{q_\Theta(\mathbf{z}_u \mid \mathbf{X}_u)}$ are intractable, thus we employ the re-parameterization trick [20] for the model optimization.

Furthermore, the diagnosis procedure $p_\Theta(\mathbf{x}_u \mid \mathbf{z}_u, \mathbf{C})$ is achieved by estimating how well a learner $u$ answers question $i$ from both the perspectives of cognitive states and comprehensive abilities. From the perspective of cognitive states, solving question $i$ requires learner $u$ to master all knowledge concepts related to this question. Regarding comprehensive abilities, each learner possesses a latent state reflecting their overall learning ability, which is shared when addressing different questions.

Formally, this process can be described as:

$$p_\Theta \left(\mathbf{x}_u \mid \mathbf{z}_u, \mathbf{C}\right) = \prod_{x_{u,i} \in \mathbf{x}_u} p_\Theta \left(x_{u,i} \mid \mathbf{z}_u, \mathbf{C}\right),$$

$$p_\Theta \left(x_{u,i} \mid \mathbf{z}_u, \mathbf{C}\right) = \sum_{c=1}^{C} c_{i,c} \cdot \phi_\Theta \left(\theta_u \cdot \mathbf{z}_u^{(c)} - \mathbf{h}_i\right), \theta_u = \sum_{c=1}^{C} \psi_\Theta \left(\mathbf{z}_u^{(c)}\right),$$

$$(3)$$

where $\phi_\Theta(\cdot)$ and $\psi_\Theta(\cdot) : \mathbb{R}^d \to \mathbb{R}^+$ are two shallow neural networks. $\psi_\Theta(\cdot)$ estimates the comprehensive ability of the learner and $\phi_\Theta(\cdot)$ predicts the performance of a learner with a given cognitive state $\mathbf{z}_u^{(c)}$ and a comprehensive ability $\theta_u$ over question $i$ in terms of concept $c$. $\mathbf{h}_i$ is a learnable latent representation for question $i$. Besides, to ensure psychometric interpretability of prediction, we set the weights of $\phi_\Theta(\cdot)$ are positive values, i.e., $\partial\phi_\Theta(\cdot)/\partial\mathbf{z}_u > 0$, assuming that the probability of correctly answering the question monotonically increases with learners' cognitive state. Please note that we found that the mean operation here can also be replaced with a neural network (i.e., $\phi'_\Theta : \mathbb{R}^C \to \mathbb{R}^+$) with positive weights, formulated as $\phi'_\Theta \left(\mathbf{c}_i \cdot \left(\theta_u \cdot \mathbf{z}_u^{(c)} - \mathbf{h}_i\right)\right)$, as in [40], without affecting prediction performance. Particularly, in contrast to most methods that consider entangled cognitive factors as input, our diagnosis model can better capture learners' proficiency on each knowledge concept by disentangling cognitive states under each concept.

Furthermore, inspired by the outstanding performance of $\beta$-TCVAE [7] in disentanglement, we prompt statistical independence among its dimensions to obtain a better trade-off between the reconstruction accuracy and the quality of disentangled representation through $q(\mathbf{z}_u^{(c)}) = \prod_{j=1}^{d} q_\Theta \left(z_{u,j}^{(c)}\right)$ where $q_\Theta(\mathbf{z}_u^{(c)})$ is the aggregated posterior of $\mathbf{z}_u$, i.e., $q_\Theta(\mathbf{z}_u^{(c)}) = \int q_\Theta(\mathbf{z}_u^{(c)} \mid \mathbf{x}_u)p(\mathbf{x}_u)\, d\mathbf{x}_u$ where $p(\mathbf{x}_u) = p_{data}(\mathbf{x}_u)$. This setup is encouraged by the term $D_{\text{KL}}(\cdot)$ in Eq. (2) based on Property 2.

**Property 2.** *The $D_{\text{KL}}(\cdot)$ in Eq. (2) can be rewritten as:*

$$D_{\text{KL}} \left(q_\Theta \left(\mathbf{z}_u \mid \mathbf{x}_u\right) \parallel p_\Theta \left(\mathbf{z}_u\right)\right) = I\left(\mathbf{z}_u, \mathbf{x}_u\right) + D_{\text{KL}} \left(q_\Theta(\mathbf{z}_u) \parallel p_\Theta(\mathbf{z}_u)\right). \tag{4}$$

See Appendix A for the proof. On one hand, $I(\mathbf{z}_u, \mathbf{x}_u)$ maximizes the mutual information (MI) between $\mathbf{z}_u$ and $\mathbf{x}_u$ which obtains the useful information for the diagnosis task as much as possible according to the information bottleneck theory [2]. On the other hand, given a Gaussian distribution $p_\Theta(\mathbf{z}_u^{(c)}) = \prod_{j=1}^{d} p_\Theta \left(z_{u,j}^{(c)}\right)$, the KL divergence term encourages independence among the dimensions of $\mathbf{z}_u^{(c)}$ by preventing each latent variable from deviating too far from specified priors. Compared to prior VAE-based CD models [50, 8], Coral additionally considers ability parameters from psychology [16] to enhance the expressive power of disentangled cognitive states.

Overall, we penalize Eq. (2) by a Lagrange multiplier $\beta$ resulting in the following objective:

$$\log p_\Theta \left(\mathbf{x}_u\right) \geq \mathbb{E}_{p(\mathbf{C})q_\Theta(\mathbf{z}_u \mid \mathbf{X}_u)} \left[\log p_\Theta \left(\mathbf{x}_u \mid \mathbf{z}_u\right)\right] - \beta \cdot \mathbb{E}_{p(\mathbf{C})} \left[D_{\text{KL}} \left(q_\Theta \left(\mathbf{z}_u \mid \mathbf{x}_u\right) \parallel p_\Theta \left(\mathbf{z}_u\right)\right)\right]. \tag{5}$$

### 3.3 Collaborative Representation Learning

Collaborative information among similar learners provides an auxiliary inter-learner insight for cognitive representation learning. However, collaborative connections among learners with similar states are typically not readily accessible. To address this challenge, we design a context-aware graph construction strategy that searches similar neighbors automatically via the initial disentangled cognitive states. Based on the constructed collaborative graph, we can learn collaborative node representations by aggregating collaborative signals to generate collaborative cognitive states.

#### 3.3.1 Context-aware Collaborative Graph Learning

The core goal of constructing the collaborative graph is to find $K$ optimal neighbors for each learner node in $V$ via their initial disentangled cognitive state $\{\mathbf{z}^{(c)}\}_{c=1}^{C}$. For different knowledge concepts, the cognitive connections between the same learner pair are typically different. Thereby, it needs to search $C$ groups of similar neighbors for each learner via each disentangled component $\mathbf{z}^{(c)}$. This means that we would generate $C$ collaborative graphs, i.e., $G = \{G_{(c)}\}_{c=1}^{C}$. Each collaborative

graph $G_{(c)}$ under concept $c$ is expected to characterize the cognitive similarities of learners regarding concept $c$. Formally, this task is defined as computing $p_\Theta(G \mid V, \mathbf{Z})$ by identifying all the $K$ similar neighbors for each learner covering each concept $c$. Let $\mathcal{N}_u^{(c)}$ denote the set of $K$ similar neighbors for the learner $u$, the task can be described as:

$$\max \log p_\Theta(G \mid V, \mathbf{Z}) := \max \sum_{c=1}^{C} \mathbb{E}_{p_\Theta\left(\mathcal{N}_u^{(c)}, \mathbf{z}_u^{(c)}\right)} \left[ \log p_\Theta\left(\mathcal{N}_u^{(c)} \mid \mathbf{z}_u^{(c)}\right) \right]$$

$$= \max \sum_{c=1}^{C} I\left(\mathcal{N}^{(c)}; \mathbf{Z}^{(c)}\right) + \sum_{c=1}^{C} \mathbb{E}_{p_\Theta\left(\mathbf{Z}^{(c)}\right)} \left[ \log p_\Theta\left(\mathbf{Z}^{(c)}\right) \right] \geq \max \sum_{c=1}^{C} I\left(\mathcal{N}^{(c)}; \mathbf{Z}^{(c)}\right), \tag{6}$$

where $\mathcal{N}^{(c)}$ and $\mathbf{Z}^{(c)} = \{\mathbf{z}_u^{(c)}\}_{u=1}^{M}$ are the neighbor set and feature set of all the learners regarding knowledge concept $c$, respectively. The number of $\mathcal{N}_u^{(c)}$ equals the combination of arbitrary $K$ neighbors from all $M$ learners for each learner node $u$ under each concept $c$, i.e., $\left|\mathcal{N}_u^{(c)}\right| = \frac{M!}{K!(M-K)!}$, thus Eq. (6) is computationally expensive especially for larger $M$ and $K$. To facilitate computation, we transform the Eq. (6) that requires global MI maximization to the task of maximizing MI locally via locally available context information inspired by [22] and derive a lower bound of it as the following Property 3.

**Property 3.** $\max \log p_\Theta(G \mid V, \mathbf{Z})$ *is bounded as follows:*

$$\max \log p_\Theta(G \mid V, \mathbf{Z}) \geq -\sum_{c=1}^{C} \sum_{u=1}^{M} \sum_{k=1}^{K} \mathcal{L}_u^{(c),k}, \text{ where } \mathcal{L}_u^{(c),k} = -\frac{\exp\left(f_{(c)}\left(b_u^{(c),k}; r_u^{(c),k-1}\right)\right)}{\sum_{v \in V_u^{(c)}} \exp\left(f_{(c)}\left(v; r_u^{(c),k-1}\right)\right)}. \tag{7}$$

See the Appendix A for the proof. The Eq. (7) iteratively searches $K$ neighbors for the learner $u$ under each knowledge concept $c$ from step $k = 1$ to $K$. $\mathcal{L}_u^{(c),k}$ is the well-known InfoNCE loss function [36]. Let $r_u^{(c),k-1}$ denote the current context at step $(k-1)$ (i.e., the set of $(k-1)$ neighbors selected from step 1 to $(k-1)$). $b_u^{(c),k}$ is the affinity candidate learner in the $(M-k)$ nonneighbor learners. Let $V_u^{(c)}$ denote the current set of nonneighbor learners, and we hence have $b_u^{(c),k} \in V_u^{(c)}$. $f_{(c)}\left(b_u^{(c),k}; r_u^{(c),k-1}\right)$ is a matching function measuring the similarity between of nonneighbor $b_u^{(c),k}$ and the current context $r_u^{(c),k-1}$, where the higher the scalar score means the higher likelihood of $b_u^{(c),k}$ is a new neighbor.

Furthermore, we have $\mathcal{L}_u^{(c),k} \propto f_{(c)}\left(b_u^{(c),k}; r_u^{(c),k-1}\right)$. Thus, given the context of $(k-1)$ neighboring learners (i.e., we have found $(k-1)$ neighbors for the learner $u$) and matching function $f_{(c)}(\cdot)$, our **goal** following the Property 3 is to find a learner $b_u^{(c),k}$ from nonneighbor set $V_u^{(c)}$ that can maximize the matching score $f_{(c)}(\cdot)$ as the $k$-th neighbor of $u$. In other words, $p(G \mid V, \mathbf{Z})$ can be optimized through maximizing the matching score $f_{(c)}(\cdot)$ from $k = 1$ to $K$ iteratively. Thereby, at each step $k$, we sort the scores of the nonneighbor learners and select the learner with the highest score to label as $k$-th neighbor $b_u^{(c),k}$, i.e., $b_u^{(c),k} \leftarrow \arg\max_v f_{(c)}(v; r_u^{(c),k-1}), v \in V_u^{(c)}$. After obtaining the $k$-th neighbor $b_u^{(c),k}$, the context $r_u^{(c),k-1}$ is updated to $r_u^{(c),k}$ by absorbing $b_u^{(c),k}$.

The calculation of matching score $f_{(c)}(\cdot)$ usually relies on the node representations (i.e., learner cognitive states). However, the sub-optimal cognitive state learning during the initial training epochs probably results in the matching function exhibiting biases. To enhance the stability of model training, instead of directly aggregating node representations as the context $r_u^{(c),k-1}$ as many graph learning works, we denote it using relative representations w.r.t. the learner $u$ [22]. Without loss of generality, we first establish relative collaborative coordinate systems with learner node $u$ as the origin, and process relationship measurements between node $u$ and each of its neighbors $v$ as $\mathbf{z}_{u,v}^{(c)} = \|\mathbf{z}_u^{(c)} - \mathbf{z}_v^{(c)}\|_2$. Then the context-aware features can be generated by aggregating each node in the context $r_u^{(c),k-1}$, i.e., $\mathbf{rc}_u^{(c),k-1} = \sum_{v \in r_u^{(c),k-1}} \mathbf{z}_{u,v}^{(c)}$. Thereby, let $v_k$ denote $b_u^{(c),k}$ with feature $\mathbf{z}_{v_k}^{(c)}$, we have $f_{(c),v_k} = f_{(c)}\left(b_u^{(c),k}; r_u^{(c),k-1}\right) = \mathbf{z}_{v_k}^{(c)\mathrm{T}} \cdot \mathbf{rc}_u^{(c),k-1}$.

### 3.3.2 Collaborative Graph Modeling

After iteratively searching $K$ neighbors under each concept, we can obtain $C$ collaborative graphs regarding each learner, i.e., $\{G_{(c)}\}_{c=1}^{C}$. Then, we consider collaborative modeling as a node representation learning task within each collaborative graph $G_{(c)}$. It relies on a nonlinear kernel function $\varphi_{\Theta}(\cdot)$ to aggregate neighboring information and update each disentangled cognitive state, i.e., $\mathbf{r}_u^{(c)} = \varphi_{\Theta}(\mathbf{z}_u^{(c)}, \{\mathbf{z}_v^{(c)} : (u,v) \in G_{(c)}\})$. Given the disentangled learner cognitive states generated by the variational posterior distribution $q_{\Theta}(\mathbf{z}_u | \mathbf{x}_u)$ from Property 1, $\varphi_{\Theta}(\cdot)$ is naturally expected to contain $C$ channels to extract different concept features from similar learners, though projecting the representation $\mathbf{z}_u$ into different subspaces, i.e., $\hat{\mathbf{z}}_u^{(c)} = \sigma(\mathbf{W}_{(c)}^{\mathrm{T}} \mathbf{z}_u^{(c)} + b_{(c)}) / \|\sigma(\mathbf{W}_{(c)}^{\mathrm{T}} \mathbf{z}_u^{(c)} + b_{(c)})\|_2$, where $\mathbf{W}_{(c)} \in \mathbb{R}^d$ and $b_{(c)} \in \mathbb{R}^d$ are learnable parameters of channel $c$ and $\sigma(\cdot)$ is a nonlinear activation function (e.g., Sigmoid), and $\|\cdot\|_2$ is $L_2$ normalization ensuring numerical stability. Then the collaborative learner representation modeling in terms of concept $c$ can be described as:

$$\mathbf{r}_u^{(c)} = \frac{1}{|\mathcal{N}_u^{(c)}|} \sum_{v \in \mathcal{N}_u^{(c)}} s_{u,v}^{(c)} \cdot \hat{\mathbf{z}}_v^{(c)}, \; s_{u,v}^{(c)} = \frac{\hat{\mathbf{z}}_u^{(c)\mathrm{T}} \cdot \hat{\mathbf{z}}_v^{(c)}}{\sum_{j \in \mathcal{N}_u^{(c)}} \hat{\mathbf{z}}_u^{(c)\mathrm{T}} \cdot \hat{\mathbf{z}}_j^{(c)}} + \frac{f_{(c),v}}{\sum_{k=1}^{K} f_{(c),v_k}}, \tag{8}$$

where $s_{u,v}^{(c)}$ is the attention weight between $u$ and $v$, considering both the collaborative aggregation (the first term) commonly used in graph modeling works and the corresponding context-aware attention (the second term) calculated in the iterative graph construction process in Eq. (7). When $K$ is set large in Eq. (7), there is a possibility of introducing non-collaborative noise. In such cases, $s_{u,v}^{(c)}$ can assign lower values to non-collaborative neighbors to mitigate the negative impact of noise, allowing for the adaptive tuning of attention in graph modeling. During training, the channels will remain changing because different subsets of the neighborhood will be searched for dynamically aggregating neighbor information in different iterations.

With Gaussian Mixture initialization from the Disentangled Cognitive Representation Encoding (section 3.2), we derive the theorem on convergence as:

**Theorem 1.** The Collaborative Representation Learning (section 3.3) procedure is equivalent to an expectation-maximization (EM) algorithm [35] for the mixture model. In particular, it converges to a point estimate of $\{\mathbf{r}_u^{(c)}\}_{c=1}^{C}$ that maximizes the marginal likelihood $l\left(\left\{a_v^{(c)} : (u,v) \in G_{(c)}\right\}_{c=1}^{C} ; \{\mathbf{r}_u^{(c)}\}_{c=1}^{C}\right)$, where $a_{u,v}^{(c)}$ equals 1 or 0 denoting whether learner $v$ is a collaborative neighbor of learner $u$ regarding concept $c$ or not. See the Appendix A for the proof.

### 3.4 Decoding and Reconstruction

Given the initial disentangled encoding via inner-learner information (section 3.2) and the collaborative representation learning via inter-learner information (section 3.3), this part encourages an alignment between the initial encode $\mathbf{z}_u$ and collaborative state $\mathbf{r}_u$, formulating a co-disentangled representation as $\tilde{\mathbf{z}}_u = \mathbf{z}_u + \mathbf{r}_u$. This operation is inspired by the residual block [17] to address the second challenge, where $\mathbf{r}_u$ can be treated as a disentangled auxiliary information of $\mathbf{z}_u$ from collaborative graphs.

The decoding process predicts the practice performance of each learner $u$ on candidate questions, given her co-disentangled representation $\tilde{\mathbf{z}}_u = \left[\tilde{\mathbf{z}}_u^{(1)}, \tilde{\mathbf{z}}_u^{(2)}, \dots, \tilde{\mathbf{z}}_u^{(C)}\right]$, i.e., $p_{\Theta}(\hat{\mathbf{x}}_u) = \mathbb{E}_{p_{\Theta}(\mathbf{C})}[p_{\Theta}(\hat{\mathbf{x}}_u \mid \tilde{\mathbf{z}}_u, \mathbf{C})]$, similar to the reconstruction procedure in Eq. (1). Thus, putting Eq. (5) and Eq. (7) together, we have the overall training objective:

$$\arg\min \mathcal{L} = \sum_{u=1}^{M} \Big[ \sum_{x_{u,i} \in \mathbf{x}_u} \alpha \cdot BCE\left(x_{u,i}, p_{\Theta}\left(x_{u,i} \mid \mathbf{z}_u, \mathbf{C}\right)\right) - \beta \cdot D_{\mathrm{KL}}^{u}$$
$$+ \sum_{x_{u,i} \in \mathbf{x}_u} BCE\left(x_{u,i}, p_{\Theta}\left(\hat{x}_{u,i}\right)\right)\Big], \tag{9}$$
$$\text{s.t.} \quad \arg\max \sum_{c=1}^{C} \sum_{k=1}^{K} \mathcal{L}_u^{(c),k},$$

| Datasets | ASSIST | Junyi | NeurIPS2020EC |
|---|---|---|---|
| #students | 1,256 | 1,400 | 1,000 |
| #questions | 16,818 | 674 | 919 |
| #knowledge concepts | 120 | 40 | 30 |
| #concepts per exercise | 1.21 | 1 | 4.02 |
| #records | 199,790 | 70,797 | 331,187 |
| #records per student | 159,07 | 50.67 | 331.19 |
| #correct records / #incorrect records | 67.08% | 77.20% | 53.87% |

Table 1: The statistics of three datasets.

where $BCE(\cdot, \cdot)$ is the binary cross entropy loss function between ground-truth practice behaviors $\mathbf{x}_u$ and the reconstructed ones. $\alpha$ and $\beta$ are hyper-parameters.

By optimizing with minimizing the above loss function Eq. (9), the cognitive state $\tilde{\mathbf{z}}_u$ of each learner $u$ can be jointly refined serving as the diagnostic results. During the testing phase, we evaluate the model performance by matching the difference between the predicted score $p_\Theta\left(\hat{x}_{u,i}\right)$ and the true score $\mathbf{x}_u$. Specifically, when a proficiency value is required instead of the vector $\tilde{\mathbf{z}}_u^{(c)}$, we can obtain it by averaging each dimension of $\tilde{\mathbf{z}}_u^{(c)}$.

# 4 Experiments

We empirically evaluate the performances of the proposed Coral model over three real-world datasets and conduct several experiments to prove its effectiveness.

## 4.1 Experimental Setup

**Datasets** We conduct experiments on three real-world datasets: ASSIST [11], Junyi [5] and NeurIPS2020EC [43]. The statistics of datasets are listed in Table 1. The details about datasets and preprocessing are depicted in the Appendix C.

**Baselines** The baselines include the matrix factorization-based model, i.e., PMF [34], the typical latent factor models derived from educational psychology, including IRT [16], MIRT [1], and the neural networks-based models, including NCDM [40], RCD [12], KaNCD [41] and DCD [8].

**Evaluation** Since cognitive states cannot be directly observed in practice, it is common to indirectly evaluate CDMs through the student performance prediction task on test datasets [4]. To evaluate prediction performance, we adopt ACC and AUC and F1-score as metrics from the perspective of classification, using a threshold of 0.5, and RMSE as metrics from the perspective of regression, following previous work [12, 23].

**Settings** We set the dimension size $d$ as 20, the layer of graph modeling as 2, and the mini-batch size as 512. In the training stage, we select the learning rate from $\{0.002, 0.005, 0.01, 0.02, 0.05\}$, select $\alpha$ from $\{0.05, 0.1, 0.5, 1\}$ and $\beta$ from $\{0.25, 0.5, 1\}$, and select neighboring number $K$ from $\{1, 2, 3, 4, 5, 10, 15, 20, 15, 30, 25, 40, 45, 50\}$. All network parameters are initialized with Xavier initialization [15]. Each model is implemented by PyTorch [37] and optimized by Adam optimizer [19]. Specially, for the implementation of baselines, we set the dimensional sizes of each representation in PMF, NCDM, KaNCD, RCD and DCD as the number of knowledge concepts. All experiments are conducted on a Linux server equipped with two 3.00GHz Intel Xeon Gold 5317 CPUs and two Tesla A100 40G GPUs.

## 4.2 Experimental Results

**Prediction Comparison** We evaluate prediction performance of Coral against baselines under three setups: normal, sparse, and cold-start scenarios.

Table 2 reports the performance comparison under normal settings for all the models across three datasets on four evaluation metrics. In this setting, we split all the datasets with a 7:1:2 ratio into training sets, validation sets, and test sets. The proposed Coral model significantly outperforms most baselines. This demonstrates two key benefits of Coral. First, the iterative graph construction process effectively generates collaborative connections for modeling. Second, the co-disentangled representation learning successfully discovers disentangled cognitive states for each learner.

We extend our analysis to assess the performance of Coral in sparse scenarios. In order to simulate varied sparse environments, we systematically discard 80%, 60%, 40%, and 20% of the training data from the ASSIST dataset under the normal settings described above. The experimental results

shown in Figure 3 (a) reveal that Coral consistently outperforms baselines across a range of sparse environments. Moreover, our model exhibits robust performance consistently, demonstrating its adaptability and effectiveness in diverse sparse scenarios.

Moreover, we conduct an analysis of Coral's performance in a cold-start environment. To replicate this scenario, we retain solely the cold-start response data for each learner in the test set of Junyi, corresponding to the knowledge concepts they had not previously practiced in the training set. The Figure 3 (b) illustrates the experimental results, highlighting the exceptional performance of the Coral model (with $K = 10$) in a cold-start scenario.

**Collaborate Graph Learning** We investigate the influence of the generated neighbor number $K$ on diagnostic performance. Figure 3 (c) displays the prediction performances for various values of $K$ on Junyi under the normal scenario. The model performance exhibits improvement as $K$ increases, particularly noticeable when $K$ is small. This observation suggests that the inter-learner information automatically retrieved by Coral contributes positively to the model. However, once $K$ surpasses a threshold, the performance gain becomes less pronounced. This diminishing effect arises because users beyond the threshold (i.e., $K = 10$ in this dataset) may lack significant collaborative relationships, thus limiting the useful clues they can offer. We observe that when $K$ exceeds the threshold, the model's

| Dataset | Method | Metric | | | |
|---|---|---|---|---|---|
| | | ACC ↑ | AUC ↑ | F1-score ↑ | RMSE ↓ |
| ASSIST | IRT | 69.36 | 69.81 | 78.14 | 45.61 |
| | MIRT | 71.26 | 72.59 | 79.80 | 44.50 |
| | PMF | 71.34 | 72.27 | 80.68 | 48.67 |
| | NCDM | 72.27 | 74.27 | 79.97 | 48.67 |
| | KaNCD | **72.43** | **75.38** | 80.22 | 48.67 |
| | RCD | 72.04 | 73.14 | 80.60 | 43.74 |
| | DCD | 70.33 | 73.98 | 79.09 | 43.94 |
| | Coral | 71.53 | 74.72 | **81.16** | **43.66** |
| Junyi | IRT | 79.26 | 76.46 | 87.54 | 38.38 |
| | MIRT | 77.74 | 74.46 | 86.05 | 40.29 |
| | PMF | 79.65 | 77.17 | 88.18 | 44.10 |
| | NCDM | 79.91 | 78.91 | 87.73 | 38.35 |
| | KaNCD | **81.79** | 80.93 | 89.02 | 36.11 |
| | RCD | 81.02 | 80.22 | 88.00 | 37.23 |
| | DCD | 79.29 | 79.55 | 87.62 | 37.83 |
| | Coral | 81.15 | **80.94** | **89.12** | **36.08** |
| NeurIPS2020EC | IRT | 70.11 | 75.60 | 71.59 | 44.68 |
| | MIRT | 69.95 | 75.52 | 71.24 | 45.51 |
| | PMF | 69.85 | 75.39 | 72.62 | 48.33 |
| | NCDM | 71.66 | 78.57 | 71.36 | 43.21 |
| | KaNCD | 71.28 | 77.60 | 72.50 | 43.71 |
| | RCD | 70.43 | 77.25 | 72.64 | 44.01 |
| | DCD | 71.53 | 75.63 | 71.13 | 45.60 |
| | Coral | **71.72** | **78.88** | **72.82** | **43.20** |

Table 2: Performance comparison. The best performance is highlighted in **bold**. ↑ (↓) means the higher (lower) score the better (worse) performance, the same as below.

performance remains acceptable, and even the performance improves after $K$ exceeds 30. This indicates that Coral effectively perceives the similarity functions of the scenario and collaborative context. Consequently, it assigns lower similarity scores to non-collaborative neighbors, robustly adjusting the attention weight in graph modeling.

To obtain a more intuitive insight into the iterative graph construction process, we randomly select two learners (called target learners) from the Junyi dataset (with the normal setup) and visualize their neighbor selection process. Initially, we utilize t-SNE [39] to present the aggregated cognitive vector of each learner $u$, which can be obtained by aggregating each disentangled cognitive component learned by Coral (setting $K = 30$), i.e., $\sum_{c=1}^{C} \tilde{\mathbf{z}}_u^{(c)}$. The embedding of the target learner is highlighted in red, while the nodes representing neighboring learners are color-coded based on the selection steps, with unselected points displayed in gray. The outcomes are illustrated in Figure 4 (a), showcasing how Coral organizes neighbors according to cognitive states and exemplifying a compelling strategy for neighbor selection that takes into account cognitive similarity.

**Disentanglement** We evaluate the disentanglement level achieved by assessing independence of dimensions within $\mathbf{z}_u$. The independence level $IL(u)$ of each $\mathbf{z}_u$ is quantified as $IL(u) = \frac{1}{C} \sum_{c=1}^{C} \frac{2}{d(d-1)} \sum_{1 \leq i,j \leq d} |z_u^{(c)}[i] - z_u^{(c)}[j]|$, where $z_u^{(c)}[i]$ represents the $i^{th}$ dimension of $\mathbf{z}_u^{(c)}$, following a prior methodology [48, 42, 49]. In Figure 3 (d), we depict $IL = \sum_{u=1}^{M} IL(u)$ and the corresponding model performances at different training epochs on ASSIST (with the normal setup). Notably, Coral (setting $K = 10$) gradually achieves a high degree of disentanglement during the training process, and the model performances generally exhibit a positive correlation with the degree of disentanglement. This observation reveals the effectiveness of the disentanglement process.

Additionally, we visualize the disentangled cognitive component representations (i.e., $\tilde{\mathbf{z}}_u^{(c)}$) of each learner $u$ learned by Coral. We treat each knowledge component in the representation as independent points, with each component colored differently. For visual clarity, we randomly select 200 learners and 5 knowledge concept components for display. The results in Figure 4 (b) uses t-SNE to visualize

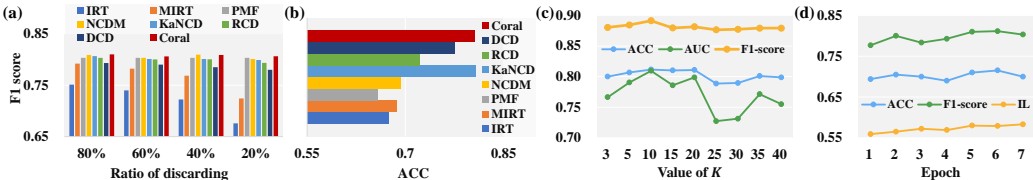

Figure 3: (a) Performance in sparse scenarios. (b) Performance under cold-start scenarios. (c) Performance with different values of $K$. (d) Disentanglement level and its correlation with performance.

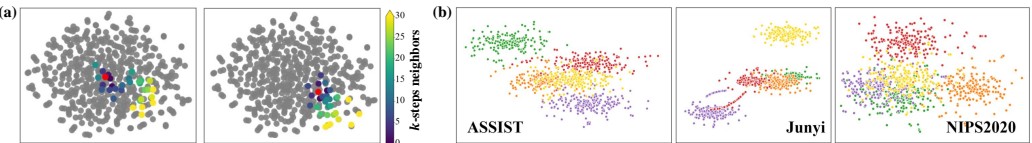

Figure 4: (a) Selected neighbors of the target learner at different steps. (b) t-SNE visualizations of learner representations colored based on knowledge concepts.

learners' cognitive states, with each color representing a distinct category of knowledge-related learner states. This illustrates Coral's ability to achieve well-separated representations.

**Explainability** We further investigate the interpretability of the cognitive diagnosis outputs based on Coral. We aim to explore whether Coral can provide reasonable predictions for knowledge concepts that learners have not practiced in the training set during actual inference. Firstly, we randomly select a target student $u$ from the Junyi dataset and identify 5 knowledge concepts (denoted as $A \sim E$) that $u$ has not learned in the training data. Subsequently, based on the refined model, we retrieve the top 4 most similar neighboring learners (i.e., $u_1 \sim u_4$) to the target student $u$. Figure 5 depicts the assessed knowledge concepts ($A \sim E$) and the corresponding mastery levels of selected neighbors using a radar chart. Table 3 presents the diagnostic outputs for the proficiency of $u$, along with 5 questions related to knowledge $A \sim E$, the predicted scores answering correctly and the actual performances of $u$. We observe that, despite the cold-start nature of these knowledge concepts for Coral, the model effectively outputs cognitive states that align with the true performance of $u$ by considering the mastery levels of collaborative learners with similar cognitive states. This sufficiently demonstrates the interpretability of Coral's diagnostic results.

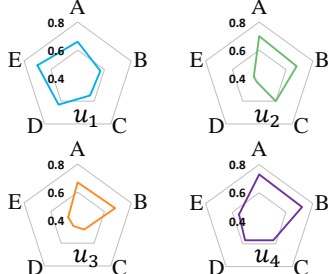

Figure 5: The example of diagnosis output.

| Question id | | 237 | 213 | 302 | 577 | 620 |
|---|---|---|---|---|---|---|
| Knowledge concept | | A | B | C | D | E |
| Proficiency (%) | | 67.2 | 68.1 | 48.3 | 52.3 | 52.6 |
| Predicted score (%) | | 73.4 | 75.2 | 47.3 | 50.8 | 57.2 |
| True performance | | ✓ | ✓ | × | × | ✓ |

Table 3: The comparison between the diagnostic results of Coral and the true performance, where ✓ denotes answering correctly and × denotes answering incorrectly.

## 5   Conclusion

We are pioneering the exploration of collaborative cognitive diagnosis by disentangling the implicit cognitive representations of learners. Extensive experiments demonstrate the superior performance of Coral, showcasing significant improvements over SOTA methods across several real-world datasets. We believe this endeavor marks a crucial step towards collaborative modeling for "AI Education". Furthermore, this work offers valuable insights into conscious-aware learner modeling, under the assumption that human learner proficiency can be effectively represented in a disentangled manner.

## Acknowledgments

This research was supported by grants from the National Key Research and Development Program of China (Grant No. 2021YFF0901003), the Key Technologies R & D Program of Anhui Province (No. 202423k09020039) and the Fundamental Research Funds for the Central Universities.

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

## A  Proofs

**Property 1.** $\max \log p_\Theta\left(\mathbf{x}_u\right)$ *is bounded as follows:*

$$\log p_\Theta\left(\mathbf{x}_u\right) \geq \mathbb{E}_{p(\mathbf{C})q_\Theta(\mathbf{z}_u|\mathbf{x}_u)}\left[\log p_\Theta\left(\mathbf{x}_u \mid \mathbf{z}_u\right)\right] - \mathbb{E}_{p(\mathbf{C})}\left[D_{\mathrm{KL}}\left(q_\Theta\left(\mathbf{z}_u \mid \mathbf{x}_u\right) \| p_\Theta\left(\mathbf{z}_u\right)\right)\right]. \quad (10)$$

The proof is as follows.

*Proof.*

$$
\begin{aligned}
\log p_\Theta\left(\mathbf{x}_u\right) &= \log \mathbb{E}_{p(\mathbf{C})}\left[p_\Theta\left(\mathbf{x}_u \mid \mathbf{z}_u, \mathbf{C}\right) p_\Theta\left(\mathbf{z}_u\right)\right] \\
&= \mathbb{E}_{p(\mathbf{C})}\left[\log p_\Theta\left(\mathbf{x}_u\right) q_\Theta\left(\mathbf{z}_u|\mathbf{x}_u\right)\right] \\
&= \mathbb{E}_{p(\mathbf{C})q_\Theta(\mathbf{z}_u|\mathbf{X}_u)}\left[\log p_\Theta\left(\mathbf{x}_u\right)\right] \\
&= \mathbb{E}_{p(\mathbf{C})q_\Theta(\mathbf{z}_u|\mathbf{X}_u)}\left[\log \frac{p_\Theta\left(\mathbf{x}_u, \mathbf{z}_u\right)}{p_\Theta\left(\mathbf{z}_u \mid \mathbf{x}_u\right)}\right] \\
&= \mathbb{E}_{p(\mathbf{C})q_\Theta(\mathbf{z}_u|\mathbf{X}_u)}\left[\log \frac{p_\Theta\left(\mathbf{x}_u, \mathbf{z}_u\right)}{p_\Theta\left(\mathbf{z}_u \mid \mathbf{x}_u\right)} \cdot \frac{q_\Theta\left(\mathbf{z}_u \mid \mathbf{x}_u\right)}{q_\Theta\left(\mathbf{z}_u \mid \mathbf{x}_u\right)}\right] \\
&= \mathbb{E}_{p(\mathbf{C})q_\Theta(\mathbf{z}_u|\mathbf{X}_u)}\left[\log \frac{q_\Theta\left(\mathbf{z}_u \mid \mathbf{x}_u\right)}{p_\Theta\left(\mathbf{z}_u \mid \mathbf{x}_u\right)}\right] + \mathbb{E}_{p(\mathbf{C})q_\Theta(\mathbf{z}_u|\mathbf{X}_u)}\left[\log \frac{p_\Theta\left(\mathbf{x}_u, \mathbf{z}_u\right)}{q_\Theta\left(\mathbf{z}_u \mid \mathbf{x}_u\right)}\right] \\
&= \mathbb{E}_{p(\mathbf{C})}\left[D_{\mathrm{KL}}\left(q_\Theta\left(\mathbf{z}_u \mid \mathbf{x}_u\right) \| p_\Theta\left(\mathbf{z}_u \mid \mathbf{x}_u\right)\right) + \mathbb{E}_{p(\mathbf{C})q_\Theta(\mathbf{z}_u|\mathbf{X}_u)}\left[\log \frac{p_\Theta\left(\mathbf{x}_u \mid \mathbf{z}_u\right) p_\Theta\left(\mathbf{z}_u\right)}{q_\Theta\left(\mathbf{z}_u \mid \mathbf{x}_u\right)}\right]\right. \\
&= \mathbb{E}_{p(\mathbf{C})}\left[D_{\mathrm{KL}}\left(q_\Theta\left(\mathbf{z}_u \mid \mathbf{x}_u\right) \| p_\Theta\left(\mathbf{z}_u \mid \mathbf{x}_u\right)\right) - D_{\mathrm{KL}}\left(q_\Theta\left(\mathbf{z}_u \mid \mathbf{x}_u\right) \| p_\Theta\left(\mathbf{z}_u\right)\right)\right. \\
&\quad + \mathbb{E}_{p(\mathbf{C})q_\Theta(\mathbf{z}_u|\mathbf{X}_u)}\left[\log p_\Theta\left(\mathbf{x}_u \mid \mathbf{z}_u\right)\right] \\
&\geq \mathbb{E}_{p(\mathbf{C})}\left[\mathbb{E}_{q_\Theta(\mathbf{z}_u|\mathbf{X}_u)}\log p_\Theta\left(\mathbf{x}_u \mid \mathbf{z}_u\right) - D_{\mathrm{KL}}\left(q_\Theta\left(\mathbf{z}_u \mid \mathbf{x}_u\right) \| p_\Theta\left(\mathbf{z}_u\right)\right)\right],
\end{aligned}
$$
$$(11)$$

which completes the proof. $\qquad\square$

**Property 2.** *The $D_{\mathrm{KL}}(\cdot)$ in Eq. (2) can be rewritten as:*

$$D_{\mathrm{KL}}\left(q_\Theta\left(\mathbf{z}_u \mid \mathbf{x}_u\right) \| p_\Theta\left(\mathbf{z}_u\right)\right)$$
$$= I\left(\mathbf{z}_u, \mathbf{x}_u\right) + D_{\mathrm{KL}}\left(q_\Theta(\mathbf{z}_u) \| p_\Theta(\mathbf{z}_u)\right). \tag{12}$$

The proof is as follows.

*Proof.* Given that $p_\Theta\left(\mathbf{x}_u\right) = p_{data}\left(\mathbf{x}_u\right)$ and $q_\Theta\left(\mathbf{z}_u, \mathbf{x}_u\right) = q_\Theta\left(\mathbf{z}_u \mid \mathbf{x}_u\right) p_\Theta\left(\mathbf{x}_u\right)$, we then have

$$D_{\mathrm{KL}}\left(q_\Theta\left(\mathbf{z}_u \mid \mathbf{x}_u\right) \| p_\Theta\left(\mathbf{z}_u\right)\right)$$
$$= \mathbb{E}_{q_\Theta(\mathbf{z}_u \mid \mathbf{x}_u)}\left[\log \frac{q_\Theta\left(\mathbf{z}_u \mid \mathbf{x}_u\right) \cdot q_\Theta\left(\mathbf{z}_u\right)}{p_\Theta\left(\mathbf{z}_u\right) \cdot q_\Theta\left(\mathbf{z}_u\right)}\right]$$
$$= \mathbb{E}_{q_\Theta(\mathbf{z}_u \mid \mathbf{x}_u)}\left[\log \frac{q_\Theta\left(\mathbf{z}_u \mid \mathbf{x}_u\right)}{q_\Theta\left(\mathbf{z}_u\right)}\right] + \mathbb{E}_{q_\Theta(\mathbf{z}_u \mid \mathbf{x}_u)}\left[\frac{q_\Theta\left(\mathbf{z}_u\right)}{p_\Theta\left(\mathbf{z}_u\right)}\right]$$
$$= \mathbb{E}_{q_\Theta(\mathbf{z}_u \mid \mathbf{x}_u) p_{data}(\mathbf{x}_u)}\left[\log \frac{q_\Theta\left(\mathbf{z}_u \mid \mathbf{x}_u\right)}{q_\Theta\left(\mathbf{z}_u\right)}\right] + \mathbb{E}_{q_\Theta(\mathbf{z}_u \mid \mathbf{x}_u) p_{data}(\mathbf{x}_u)}\left[\frac{q_\Theta\left(\mathbf{z}_u\right)}{p_\Theta\left(\mathbf{z}_u\right)}\right] \tag{13}$$
$$= \mathbb{E}_{q_\Theta(\mathbf{z}_u, \mathbf{x}_u)}\left[\log \frac{q_\Theta\left(\mathbf{z}_u \mid \mathbf{x}_u\right)}{q_\Theta\left(\mathbf{z}_u\right)}\right] + \mathbb{E}_{q_\Theta(\mathbf{z}_u)}\left[\frac{q_\Theta\left(\mathbf{z}_u\right)}{p_\Theta\left(\mathbf{z}_u\right)}\right]$$
$$= I\left(\mathbf{z}_u; \mathbf{x}_u\right) + D_{\mathrm{KL}}\left(q_\Theta\left(\mathbf{z}_u\right) \| p_\Theta\left(\mathbf{z}_u\right)\right),$$

where $I(\mathbf{A}; \mathbf{B})$ calculates mutual information (MI) between $\mathbf{A}$ and $\mathbf{B}$, i.e. $I(\mathbf{A}; \mathbf{B}) = \mathbb{E}_{p(\mathbf{a}, \mathbf{b})}[\log \frac{p(\mathbf{a}|\mathbf{b})}{p(\mathbf{a})}]$. Therefore, the proof is completed. $\square$

**Property 3.** $\max \log p_\Theta(G \mid V, \mathbf{Z})$ *is bounded as follows:*

$$\max \log p_\Theta(G \mid V, \mathbf{Z}) \geq -\sum_{c=1}^{C}\sum_{u=1}^{M}\sum_{k=1}^{K} \mathcal{L}_u^{(c),k}$$
$$\text{where } \mathcal{L}_u^{(c),k} = -\frac{\exp\left(f_{(c)}\left(b_u^{(c),k}; r_u^{(c),k-1}\right)\right)}{\sum_{v \in V_u^{(c)}} \exp\left(f_{(c)}\left(v; r_u^{(c),k-1}\right)\right)}. \tag{14}$$

The proof is as follows:

*Proof.* Given the following inequality,

$$\max \log p_\Theta(G \mid V, \mathbf{Z}) := \max \sum_{c=1}^{C} \mathbb{E}_{p_\Theta\left(\mathcal{N}_u^{(c)}, \mathbf{z}_u^{(c)}\right)}\left[\log p_\Theta\left(\mathcal{N}_u^{(c)} \mid \mathbf{z}_u^{(c)}\right)\right]$$
$$= \max \sum_{c=1}^{C} I\left(\mathcal{N}^{(c)}; \mathbf{Z}^{(c)}\right) + \sum_{c=1}^{C} \mathbb{E}_{p_\Theta(\mathbf{Z}^{(c)})}\left[\log p_\Theta\left(\mathbf{Z}^{(c)}\right)\right] \tag{15}$$
$$\geq \max \sum_{c=1}^{C} I\left(\mathcal{N}^{(c)}; \mathbf{Z}^{(c)}\right),$$

we further derive the term $I\left(\mathcal{N}^{(c)}; \mathbf{Z}^{(c)}\right)$ that search the combination of $K$ similar neighbors from the permutation perspective as follows:

$$I\left(\mathcal{N}^{(c)}; \mathbf{Z}^{(c)}\right) = \mathbb{E}_{p_\Theta\left(\mathcal{N}_u^{(c)}, \mathbf{z}_u^{(c)}\right)}\left[\log p_\Theta\left(\mathcal{N}_u^{(c)} \mid \mathbf{z}_u^{(c)}\right)\right] + H\left(\mathcal{N}_u^{(c)}\right)$$
$$\geq \mathbb{E}_{p\left(\mathcal{R}_u^{(c)}, \mathbf{z}_u^{(c)}\right)}\left[\log p_\Theta\left(\mathcal{R}_u^{(c)} \mid \mathbf{z}_u^{(c)}\right)\right] + H\left(\mathcal{N}_u^{(c)}\right), \tag{16}$$

where $H(\mathbf{A}) = -\sum p(\mathbf{a}) \cdot \log p(\mathbf{a})$. $\mathcal{R}_u^{(c)}$ denotes the permutation of neighbors representing routes to $\mathcal{N}_u^{(c)}$ in given orders, where $\left|\mathcal{R}_u^{(c)}\right| = \frac{M!}{(M-K)!}$. However, the search space of $\mathcal{R}_u^{(c)}$ is huge and even prohibitive. Inspired by the equivalent task [22], we next present the Eq. (16) in a heuristic

style by maximizing the MI between the context of selected similar learner neighbors and the next neighbor iteratively.

Specifically, the core goal of Eq. (15) is to find all $K$ neighbors $\mathcal{N}_u^{(c)}$ for each learner $u$ under a specific concept the data at once, from the perspective of maximizing MI globally. However, this task is challenging due to the especially large search spaces of $\mathcal{N}_u^{(c)}$ and $\mathcal{R}_u^{(c)}$. Thereby, we decompose the globally optimal task into an equivalent task in an iterative local optimal process. Concretely, assume that we have found $(k_0 - 1)$ optimal neighbors for the learner $u$ formulating a route $r_u^{(c),k_0-1}$ in a specific order from learner node 1 to $(k_0 - 1)$, we then search the $k_0$-th neighbor $b_u^{(c),k_0}$ equally from the rest $(M - k_0 + 1)$ learners for the learner $u$, i.e., $p\left(b_u^{(c),k_0}\right) = \frac{1}{M-k_0+1}$.

Given arbitrary $k_0$ optimal neighbors for learner $u$ with a specific sub-route $r_u^{(c),k_0}$, we can derive $p_\Theta\left(\mathcal{R}_u^{(c)}, \mathbf{z}_u^{(c)}\right)$ in Eq. (16) as follows:

$$p_\Theta\left(\mathcal{R}_u^{(c)}, \mathbf{z}_u^{(c)}\right) = \mathbb{E}_{p_\Theta\left(\mathcal{R}_u^{(c)}\right)}\left[p_\Theta\left(r_u^{(c),k_0}\right)\prod_{i=k_0+1}^{K} p_\Theta\left(b_u^{(c),i} \mid r_u^{(c),i-1}\right)\right]. \qquad (17)$$

We can derive the log term $\log p_\Theta\left(\mathcal{R}_u^{(c)} \mid \mathbf{z}_u^{(c)}\right)$ in Eq. (16) as follows:

$$
\begin{aligned}
\log p_\Theta\left(\mathcal{R}_u^{(c)} \mid \mathbf{z}_u^{(c)}\right) &= \mathbb{E}_{p_\Theta\left(\mathcal{R}_u^{(c)}\right)}\left[\sum_{k=1}^{K} \log p_\Theta\left(b_u^{(c),k} \mid r_u^{(c),k-1}\right)\right] \\
&= \mathbb{E}_{p_\Theta\left(\mathcal{R}_u^{(c)}\right)}\left[\sum_{k=1}^{K} \log \frac{p_\Theta\left(b_u^{(c),k} \mid r_u^{(c),k-1}\right) \cdot p_\Theta\left(b_u^{(c),k}\right)}{p_\Theta\left(b_u^{(c),k}\right)}\right] \\
&= \mathbb{E}_{p_\Theta\left(\mathcal{R}_u^{(c)}\right)}\left[\sum_{k=1}^{K} \log \frac{p_\Theta\left(b_u^{(c),k} \mid r_u^{(c),k-1}\right)}{p_\Theta\left(b_u^{(c),k}\right)} + \sum_{k=1}^{K} \log p_\Theta\left(b_u^{(c),k}\right)\right].
\end{aligned}
\qquad (18)
$$

With Eq. (17) and (18), we can derive the first term in Eq. (16) as follows:

$$
\begin{aligned}
&\mathbb{E}_{p_\Theta\left(\mathcal{R}_u^{(c)}, \mathbf{z}_u^{(c)}\right)}\left[\log p_\Theta\left(\mathcal{R}_u^{(c)} \mid \mathbf{z}_u^{(c)}\right)\right] \\
&= \sum_{u=1}^{M} \mathbb{E}_{p_\Theta\left(\mathcal{R}_u^{(c)}\right)}\left[p_\Theta\left(r_u^{(c),k_0}\right)\prod_{i=k_0+1}^{K} p_\Theta\left(b_u^{(c),i} \mid r_u^{(c),i-1}\right)\sum_{k=1}^{K} \log \frac{p_\Theta\left(b_u^{(c),k} \mid r_u^{(c),k-1}\right)}{p_\Theta\left(b_u^{(c),k}\right)}\right] \\
&\quad + \sum_{u=1}^{M} \mathbb{E}_{p_\Theta\left(\mathcal{R}_u^{(c)}\right)}\left[p_\Theta\left(r_u^{(c),k_0}\right)\prod_{i=k_0+1}^{K} p_\Theta\left(b_u^{(c),i} \mid r_u^{(c),i-1}\right)\sum_{k=1}^{K} \log p_\Theta\left(b_u^{(c),k}\right)\right] \\
&\approx \sum_{u=1}^{M} \mathbb{E}_{p_\Theta\left(\mathcal{R}_u^{(c)}\right)}\left\{\sum_{k=1}^{K}\left[I\left(b_u^{(c),k}; r_u^{(c),k-1}\right)\prod_{i=k}^{K} p_\Theta\left(b_u^{(c),i} \mid r_u^{(c),i-1}\right)\right]\right\} + \epsilon(M, K),
\end{aligned}
\qquad (19)
$$

where $\epsilon(M, K) \geq 0$ is a constant term regarding $M$ and $K$. Inspired by [36], we have the lower bound of $I\left(b_u^{(c),k}; r_u^{(c),k-1}\right)$ as follows:

$$I\left(b_u^{(c),k}; r_u^{(c),k-1}\right) \geq \log M - \mathcal{L}_u^{(c),k},$$

$$\text{where} \quad \mathcal{L}_u^{(c),k} = -\frac{\exp\left(f_{(c)}\left(b_u^{(c),k}; r_u^{(c),k-1}\right)\right)}{\sum_{v \in V_u^{(c)}} \exp\left(f_{(c)}\left(v; r_u^{(c),k-1}\right)\right)}. \qquad (20)$$

The Eq. (20) iteratively searches $K$ neighbors for the learner $u$ under each knowledge concept $c$ from step $k = 1$ to $K$. $\mathcal{L}_u^{(c),k}$ is the well-known InfoNCE loss function [36]. Let $r_u^{(c),k-1}$ denote the

current context at step $(k-1)$ (i.e., the set of $(k-1)$ neighbors selected from step 1 to $(k-1)$). $b_u^{(c),k}$ is the affinity candidate learner in the $(M-k)$ nonneighbor learners. Let $V_u^{(c)}$ denote the current set of nonneighbor learners, and we hence have $b_u^{(c),k} \in V_u^{(c)}$. $f_{(c)}\left(b_u^{(c),k}; r_u^{(c),k-1}\right)$ is a matching function measuring the similarity between of nonneighbor $b_u^{(c),k}$ and the current context $r_u^{(c),k-1}$, where the higher the scalar score means the higher likelihood of $b_u^{(c),k}$ is a new neighbor.

Furthermore, we have $\mathcal{L}_u^{(c),k} \propto f_{(c)}\left(b_u^{(c),k}; r_u^{(c),k-1}\right)$. Thus, given the context of $(k-1)$ neighboring learners (i.e., we have found $(k-1)$ neighbors for the learner $u$) and matching function $f_{(c)}(\cdot)$, our goal following the Eq. (20) is to find a learner $b_u^{(c),k}$ from nonneighbor set $V_u^{(c)}$ that can maximize the matching score $f_{(c)}(\cdot)$ as the $k$-th neighbor of $u$. In other words, $p(G \mid V, \mathbf{Z})$ can be optimized through maximizing the matching score $f_{(c)}(\cdot)$ from $k=1$ to $K$ iteratively. Thereby, at each step $k$, we sort the scores of the nonneighbor learners and select the learner with the highest score to label as $k$-th neighbor $b_u^{(c),k}$, i.e., $b_u^{(c),k} \leftarrow \arg\max_v f_{(c)}(v; r_u^{(c),k-1}), v \in V_u^{(c)}$. After obtaining the $k$-th neighbor $b_u^{(c),k}$, the context $r_u^{(c),k-1}$ is updated to $r_u^{(c),k}$ by absorbing $b_u^{(c),k}$.

Based on the Eq. (15), Eq. (16) and Eq. (20), we have

$$\max \log p(G \mid V, \mathbf{Z}) \geq -\sum_{c=1}^{C} \sum_{u=1}^{M} \sum_{k=1}^{K} \mathcal{L}_u^{(c),k}, \tag{21}$$

which completes the proof.

$\square$

**Theorem 1.** With Gaussian Mixture initialization from the Disentangled Cognitive Representation Encoding (section 3.2), the Collaborative Representation Learning (section 3.3) procedure is equivalent to an expectation-maximization (EM) algorithm [35] for the mixture model. In particular, it converges to a point estimate of $\{\mathbf{r}_u^{(c)}\}_{c=1}^{C}$ that maximizes the marginal likelihood $l\left(\left\{a_v^{(c)} : (u,v) \in G_{(c)}\right\}_{c=1}^{C}; \{\mathbf{r}_u^{(c)}\}_{c=1}^{C}\right)$, where $a_{u,v}^{(c)}$ equals 1 or 0 denoting whether learner $v$ is a collaborative neighbor of learner $u$ regarding concept $c$ or not.

The proof is as follows.

*Proof.* The collaborative modeling process can be approximatively equivalent to an expectation-maximization (EM) algorithm for the mixture model. Let $A = \{A_{(c)}\}_{c=1}^{C}$ where $A_{(c)} = \{a_v^{(c)} : (u,v) \in G_{(c)}\}$, where $a_{u,v}^{(c)}$ equals 1 or 0 denoting whether learner $v$ is a collaborative neighbor of learner $u$ regarding concept $c$ or not, which is a type unknown factor in EM algorithm. Given $\mathbf{R} = \{\mathbf{r}^{(c)}\}_{c=1}^{C}$ and $\mathbf{Z} = \{\mathbf{z}^{(c)}\}_{c=1}^{C}$, the EM algorithm maximizes the likelihood $l(A; \mathbf{R}) = \sum_A l(A, \mathbf{Z}; \mathbf{R})$. Let $q(A)$ is the distribution over $A$, we then have

$$\begin{aligned}
\log l(A; \mathbf{R}) &= \sum_A q(A) \cdot l(\mathbf{Z}; \mathbf{R}) \\
&= \sum_A q(A) \cdot \frac{l(A, \mathbf{Z}; \mathbf{R})}{l(A \mid \mathbf{Z}; \mathbf{R})} \\
&= \sum_A q(A) \cdot \frac{l(A, \mathbf{Z}; \mathbf{R})}{q(A)} + \sum_A q(A) \cdot \frac{q(A)}{l(A \mid \mathbf{Z}; \mathbf{R})}.
\end{aligned} \tag{22}$$

Let $L(\mathbf{R}, q(A))$ denote $\sum_A q(A) \cdot \frac{l(A, \mathbf{Z}; \mathbf{R})}{q(A)}$, we can rewrite Eq. (22) as

$$\begin{aligned}
\log l(\mathbf{Z}; \mathbf{R}) &= L(\mathbf{R}, q(A)) + D_{\mathrm{KL}}(q(A) \parallel l(A \mid \mathbf{Z}; \mathbf{R})) \\
&\leq L(\mathbf{R}, q(A)),
\end{aligned} \tag{23}$$

where $L(\mathbf{R}, q(A))$ is a lower bound of $l(A; \mathbf{R})$ since the KL divergence from $l(A \mid \mathbf{Z}; \mathbf{R})$ towards $q(A)$ is non-negative.

At every E-step, the EM algorithm aims to search the optimal $q(A)$ that tightens the lower bound, which $q(A)$ is set to $l(A \mid \mathbf{Z}; \mathbf{R})$ since the KL divergence will become zero. Given that

$$l(A \mid \mathbf{Z}; \mathbf{R}) = \log p(G \mid V, \mathbf{Z}), \tag{24}$$

and Property 3, the optimal $q(A)$ that tightens the lower bound can be achieved through iteratively maximizing the following objective as

$$\max \sum_{c=1}^{C} \sum_{u=1}^{M} \sum_{k=1}^{K} \frac{\exp\left(f_{(c)}\left(b_u^{(c),k}; r_u^{(c),k-1}\right)\right)}{\sum_{v \in V_u^{(c)}} \exp\left(f_{(c)}\left(v; r_u^{(c),k-1}\right)\right)}, \tag{25}$$

which performs the E-step.

After the E-step, an M-step is performed to maximize the lower bound $L(\mathbf{R}, q(A))$ w.r.t. $\mathbf{R}$ with $q(A)$ fixed found in the E-step. Given that

$$\mathbf{r}_u^{(c)} = \frac{1}{|\mathcal{N}_u^{(c)}|} \sum_{v \in \mathcal{N}_u^{(c)}} s_{u,v}^{(c)} \cdot \mathbf{z}_v^{(c)}, \tag{26}$$

we optimize the $\mathbf{r}^{(c)}$ upon $\frac{\partial L(\mathbf{R}, q(A))}{\partial \mathbf{r}^{(c)}}$ to zero, which is actually performing the M-step.

Let $q(A)^k$ and $\mathbf{r}^k$ be the result of the $k$-th E-step and the $k$-th M-step, respectively, we then have

$$
\begin{aligned}
l\left(A^k; \mathbf{R}^{k-1}\right) &= l\left(\mathbf{R}^{k-1}, q(A)^k\right) + D_{\mathrm{KL}}\left(q(A)^k \parallel l\left(A^k \mid \mathbf{Z}; \mathbf{R}^{k-1}\right)\right) \\
&= L\left(\mathbf{R}^{k-1}, q(A)^k\right) \\
&\leq L\left(\mathbf{R}^k, q(A)^k\right) \\
&\leq L\left(\mathbf{R}^k, q(A)^k\right) + D_{\mathrm{KL}}\left(q(A)^k \parallel l\left(A \mid \mathbf{Z}; \mathbf{R}^k\right)\right) \\
&= l\left(A^k; \mathbf{R}^k\right).
\end{aligned}
\tag{27}
$$

Hence, this can prove that the likelihood will increase monotonically and be upper-bounded by zero at the same time. Therefore, the algorithm converges, which completes the proof. □

# B  Algorithm

To offer a more comprehensive description of Coral's structure, we outline the algorithm (see Algorithm 1) for the entire model, including four functions and a main function.

# C  Dataset Description and Preprocessing

We conduct experiments on three real-world datasets, i.e., ASSIST [11], Junyi [5] and NeurIPS2020EC [43]. The statistics of these datasets are summarized in Table 1. For all datasets, we preserve the first-time exercise-answering record for the same learner-question pairs to support cognitive diagnosis aligning with common settings used in previous related studies [40]. The detailed information on datasets and preprocessing method are depicted as follows:

- **ASSIST (ASSISTments 2009-2010 "skill builder")** [11] This dataset is an open dataset collected by the ASSISTments online tutoring systems[2], which has become one of the popular benchmark datasets for cognitive diagnosis. We preserve learners with more than 30 practice records for ASSIST to guarantee that each learner has enough data for diagnosis.

- **Junyi** [5] This dataset contains learner online learning logs collected from a Chinese online educational platform called Junyi Academy[3]. Nowadays, Junyi is widely used in the evaluation of online education tasks [47, 8]. We randomly select 1,400 learners with more than 15 practice records from Junyi to guarantee that each learner has enough data for diagnosis.

---

[2]https://sites.google.com/site/assistmentsdata/
[3]https://www.junyiacademy.org/

---

**Algorithm 1** Coral Model

---

1: **Input:** $\{\mathbf{c}\}_{i=1}^{N}$, practice logs $\{\mathbf{x}_u\}_{u=1}^{M}$;
2: **function** DISENTANGLED_COGNITIVE_REPRESENTATION_ENCODING($\{\mathbf{c}\}_{i=1}^{N}, \mathbf{x}_u$)
3:   // Gaussian Mixture initialization of learner $u$
4:   $\mathbf{z}_u \leftarrow [\mathbf{z}_u^{(1)}; \mathbf{z}_u^{(2)}; \ldots; \mathbf{z}_u^{(C)}]$
5:   // Calculate ability of learner $u$
6:   $\theta_u \leftarrow \psi_\Theta(\mathbf{z}_u^{(c)}), c = 1, 2, \ldots, C$
7:   // Reconstruct practice performance of learner $u$
8:   $p_\Theta(x_{u,i} \mid \cdot) \leftarrow c_{i,c} \cdot \phi_\Theta(\theta_u, \mathbf{z}_u^{(c)}), c = 1, 2, \ldots, C$
9:   **return** $BCE(x_{u,i}, p_\Theta(x_{u,i} \mid \cdot)), D_{\text{KL}}^u, \mathbf{z}_u$
10: **end function**

11: // Search $K$ neighbors for learner $u$ from all the learners $V$
12: **function** CONTEXT-AWARE_COLLABORATIVE_GRAPH_LEARNING($V, \mathbf{z}_u^{(c)}$)
13:   **for** $c = 1, 2, \ldots, C$ **do**
14:     // Let  be the removing operation
15:     $V_u^{(c)} \leftarrow V \setminus u$
16:     // Initial neighbor set of $u$, i.e., $R_u$
17:     $r_u^{(c),k} \leftarrow \{u\}$, where $k = 0$
18:     // Iteratively calculating the cognitive similarity scores between $u$ and each learner in $V_u^{(c)}$, the initial step corresponds to $k = 0$
19:     **for** $k = 1, \ldots, K$ **do**
20:       **for** $v \in V_u^{(c)}$ **do**
21:         $Score_{u,v} \leftarrow f_{(c)}(v; r_u^{(c),k-1}) // Equation(7)$
22:       **end for**
23:       // Select $k$-th neighbor for $u$, denoted as $b_u^{(c),k}$
24:       $b_u^{(c),k} \leftarrow \arg\max_{v} Score_{u,v}, v \in V_u^{(c)}$
25:       // Update neighbors and non-neighbor learners
26:       $r_u^{(c),k} \leftarrow r_u^{(c),k-1} + \{b_u^{(c),k}\}$
27:       $V_u^{(c)} \leftarrow V_u^{(c)} \setminus b_u^{(c),k}$
28:     **end for**
29:   **end for**
30:   **return** $\{G_{(c)}\}_{c=1}^{C}$
31: **end function**

32: **function** COLLABORATIVE_GRAPH_MODELING($\{G_{(c)}\}_{c=1}^{C}, \mathbf{z}_u$)
33:   **for** $c = 1, 2, \ldots, C$ **do**
34:     $\mathbf{r}_u \leftarrow \varphi(\mathbf{Z}, G)$
35:   **end for**
36:   **return** $\mathbf{r}_u$
37: **end function**

38: **function** DECODING_AND_RECONSTRUCTION($\{G_{(c)}\}_{c=1}^{C}, \mathbf{z}_u, \mathbf{x}_u$)
39:   Calculate $\tilde{\mathbf{z}}_u$
40:   **return** $BCE(x_{u,i}, p_\Theta(\hat{x}_{u,i}))$
41: **end function**

42: **BEGIN MAIN FUNCTION:**
43: **Initialize** $\mathbf{z}_u$, learning rate $lr$, $\beta$, $epoch \leftarrow 0, TotalEpoch$
44: **repeat**
45:   **for** $u = 1, 2, \ldots, M$ **do**
46:     $BCE(x_{u,i}, p_\Theta(x_{u,i} \mid \cdot)), D_{\text{KL}}^u, \mathbf{z}_u \leftarrow$ Disentangled_Cognitive_Representation_Encoding($\ldots$) // $\ldots$ denotes omitting parameters
47:     $\{G_{(c)}\}_{c=1}^{C} \leftarrow$ Context-aware_Collaborative_Graph_Learning($\ldots$)
48:     $BCE(x_{u,i}, p_\Theta(\hat{x}_{u,i})) \leftarrow$ Decoding_and_Reconstruction($\ldots$)
49:     Calculate $\arg\min \mathcal{L}$
50:     $\Theta \leftarrow \arg\max_\Theta \mathcal{L}$ by $lr \cdot \nabla_\Theta \mathcal{L}$
51:     $epoch \leftarrow epoch + 1$
52:   **end for**
53: **until** $epoch$ equals $TotalEpoch$

---

- **NeurIPS2020EC** [43] This dataset is originated from NeurIPS 2020 Education Challenge, which provides learners' practice logs on mathematical questions from Eedi[4]. We randomly select 1,000 learners with more than 15 practice records from NeurIPS2020EC to guarantee that each learner has enough data for diagnosis.

# D  Additional Experimental Results

| Method | Metric | | | |
|---|---|---|---|---|
| | ACC ↑ | AUC ↑ | F1-score ↑ | RMSE ↓ |
| w/o KL | 0.693156 | 0.659914 | 0.803728 | 0.452067 |
| w/o collar | 0.667321 | 0.606090 | 0.786824 | 0.465231 |
| w/ knn | 0.708520 | 0.721339 | 0.794198 | 0.440430 |
| Coral | **0.709710** | **0.721823** | **0.810755** | **0.437818** |

Table 4: Ablation study of Coral on ASSIST.

## D.1  Ablation Study

We additionally perform ablation studies to assess the impact of key components within Coral. The results in Table 4 depict the performances of Coral (setting $K = 5$) under various conditions: without the KL term for encoding (w/o KL), without the collaborative aggregation during decoding (w/o collar), and replacing the collaborative graph construction procedure using a knn-based methods (w/ knn) used in [13] on the ASSIST dataset. These findings show the effectiveness of each key component in enhancing the overall performance of Coral.

## D.2  Efficiency Improvement

We additionally implement three efficiency optimization strategies to further reduce the complexity of Coral. These strategies cannot theoretically guarantee optimal performance, but they can enhance applicability and scalability of Coral through empirical balancing of efficiency and accuracy. We refer them as Coral with $n$-sample, Coral with $m$-selections, and Coral with full-kit, as follows:

- Coral with $n$-sample: During the $K$ iterations of searching for neighbors, randomly sample $n$ subsets from all $M$ learners to replace $V$ in the original approach. This reduces computational efficiency from $M \times K$ to $n \times K$, where $n \ll M$.

- Coral with $m$-selections: Based on the basic Coral, replace selecting one neighbor per iteration with selecting $m$ neighbors. This decreases computational efficiency from $M \times K$ to $\frac{M \times K}{m}$, where $m < K$.

- Coral with full-kit: A combination of Coral with $n$-sample and Coral with $m$-selections, further reducing computational efficiency from $M \times K$ to $\frac{n \times K}{m}$, where $n \ll M$ and $m < K$.

Following the three strategies outlined above, we conduct several experiments on the Junyi dataset, with $K = 40$, to assess prediction performance. The results are summarized in the Figure 6. These experimental results demonstrate Coral's potential to improve computational efficiency while maintaining acceptable performance, as evidenced by the varying levels of accuracy achieved with different optimization strategies.

# E  Broader Impact and Limitation

This research delves into modeling human cognitive states within the realm of intelligent education. The proposed Coral model significantly enhances the diagnostic accuracy of implicit learners' knowledge states. This improvement not only provides effective insights for online personalized tutoring services, such as question recommendations but also lays the foundation for further research in this area. Moreover, the automatic construction strategy for collaborative connections among learners

---

[4]https://eedi.com/

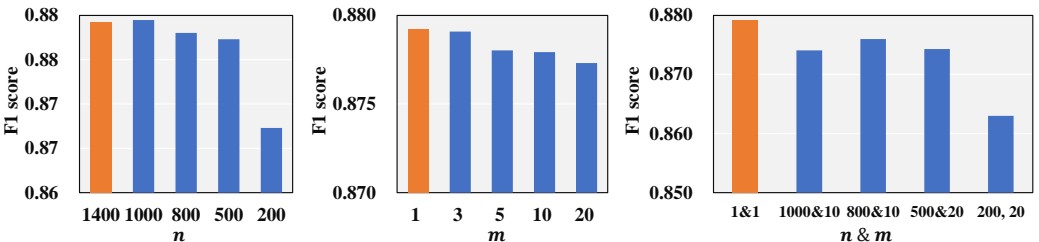

Figure 6: The prediction performance of the improved model is illustrated, with the orange bar representing the performance of the original Coral, which achieves the highest F1 score.

offers valuable insights that can contribute to subsequent investigations in this field. Lastly, we anticipate that the proposed techniques can be extended to other domains, including but not limited to user interest modeling and social network modeling. Although our method is effective both theoretically and empirically, it suffers from computational inefficiencies. We have explored preliminary optimization strategies in the Appendix D.2 and will focus on improving computational efficiency in future research. In addition, future research plans to consider issues of fairness [51, 52] and explore the integration of large language models and multi-modal knowledge to enhance interpretability [27, 29, 30]. In essence, our work is dedicated to advancing intelligent education and deepening the understanding of human cognitive proficiency. It cannot cause negative effects. We anticipate its crucial role in fostering progress in both pertinent technologies and societal advancements.

