# OpenReview forum: "Collaborative Cognitive Diagnosis with Disentangled Representation Learning for Learner Modeling"
_NeurIPS.cc/2024/Conference — NeurIPS 2024 poster_

### Official Review · Reviewer_Ty5J · 2024-07-10

**Soundness:** 3
**Presentation:** 3
**Contribution:** 2
**Rating:** 5
**Confidence:** 3

**Summary:**

This paper introduces Coral, a model that enhances cognitive diagnosis by integrating collaborative signals among learners with disentangled representation learning. Coral addresses the importance of leveraging collaborative connections among learners to better understand human learning processes. Traditional cognitive diagnosis methods focus on individual attributions and explicit practice records, but they often overlook the potential of collaborative signals. The proposed Coral model aims to fill this gap by integrating collaborative and disentangled cognitive states. The disentangled state encoder separates the cognitive states of learners and reconstructs their practice performance from an inner-learner perspective. The collaborative representation learning dynamically constructs a collaborative graph by iteratively searching for optimal neighbors in a context-aware manner, effectively capturing collaborative signals. The model is evaluated on three real-world datasets: ASSIST, Junyi, and NIPS2020EC. The results show that Coral outperforms existing methods in terms of ACC, AUC, F1-score, and RMSE. Coral also performs well in sparse and cold-start scenarios, demonstrating its robustness and adaptability.

**Strengths:**

1. Coral introduces an integration of collaborative signals with disentangled representation learning, addressing a gap in existing cognitive diagnosis methods.
2. Experimental results demonstrate performance improvements over state-of-the-art methods across various datasets and scenarios.
3. The disentangled representation provides better explainability and controllability of cognitive states, which is crucial for personalized education.

**Weaknesses:**

1. The approach in this paper is not very innovative, and the motivation is not very convincing.
2. Although visualizing the embeddings provides some level of interpretability, it still does not yield any explicit conclusions or observable phenomena. Rather than visualizing the embeddings (t-SNE has a certain degree of randomness), I would prefer to see some interesting conclusions based on the dataset.
3. The paper mentions computational inefficiencies, which could limit the model's applicability in large-scale or real-time settings.
4. The model requires careful tuning of hyperparameters, which may not be straightforward for all datasets and might require extensive experimentation.
5. The paper could benefit from a more extensive comparison with a broader range of alternative approaches to strengthen its contributions.

**Questions:**

Please see the weaknesses

---

> ### Author Rebuttal · Authors · 2024-08-06
>
> We appreciate your careful reading and detailed feedback on our paper. We address your concerns below and please let us know if there are remaining questions or unclear points.
> > **Weakness 1:** Unconvincing motivation and limited methodological novelty.
>
> Thank you for highlighting these concerns and for giving us the opportunity to provide a clearer explanation.
> - **Our motivation is well-founded** and supported by data analysis and relevant literature, involving two aspects:
> 1. Motivation 1 (collaborative modeling): Introducing collaborative signals between learners in CD can enhance performance and interpretability. This is supported by social learning theories[1], data analysis (see experiments 1 and 2 in the next response to Weakness 2), and empirical effectiveness on learner sequences[2] and group modeling[3], which demonstrate the benefits of incorporating collaborative signals. However, most CD models focus on individual modeling and often neglect collaborative aspects. Thus, we incorporate collaborative signals in to CD is both convincing and meaningful.
> 2. Motivation 2 (disentanglement): Modeling both individual and collaborative learner representations following Motivation 1 increases complexity, which traditional single-vector representations may not adequately handle. Thus, disentanglement is naturally introduced to convert single-vector representations into multiple vectors, reducing dimensional correlation and enhancing modeling ability, inspired by [4].
>
> - **Mthodological novelty** on key modules:
> 1. Disentangled Encoder (inner-learner view): Inspired by DCD[3], our encoder considers both vector-level and dimensional-level disentanglement, unlike DCD, which only considers the latter. This approach's effectiveness is empirically demonstrated through ablation studies.
> 2. Collaborative Modeling (inter-learner view): This design is unique as relevant work in CD is almost blank. We provide theoretical derivations to support the model's rationale.
> 3. Co-Disentanglement Decoder:  Inspired by residual connections, our decoder effectively and simply aligns inner- and inter-learner perspectives. This alignment represents a novel exploration in this direction.
>
> Our model's innovation is supported by both empirical and theoretical evidence.
>
> > **Weakness 2:** Concerns on interpretability.
>
> Thank you for your attention to interpretability.
>
> We would like to clarify that our interpretability analysis is detailed in a case study in Appendix E, rather than through visualizations. This example demonstrates how Coral infers each learner's future performance by referencing the cognitive states of similar learners, thereby providing educators with a basis for their inferences.
>
> To provide more data-based conclusions, we conduct additional experiments on ASSIST:
> 1. We represent $M$ learners $U$’ performance on $N$ questions with a matrix $X = \\{x\_{u,i}\\}\_{M \times N}$, where $x\_{u,i}$ indicates whether a learner $u$ answered problem $i$ correctly (1), incorrectly (0), or did not attempt it (-1). We split this matrix into $X\_{\text{train}} = \\{x\_{u,i} \mid u\in U, i \in [1, N/2]\\}$ and $X\_{\text{test}} = \\{x\_{u,i} \mid u\in U,i \in [N/2+1, N]\\}$. For each learner $u$ in $X\_{\text{train}}$, we find the most similar learner $v$ by maximizing $\text{cosine}(X\_{u}, X\_{v})$, and then compute the absolute difference $|\Delta(u,v)|$ in their passing rates on $X\_{\text{test}}$.
> 2. Based on a trained Coral, we find the most similar learner $v$ for each learner $u$ via learned embedding cosine similarity. We then calculate the absolute difference $|\Delta(u,v)|$ in the accuracy of problem-solving between $u$ and $v$ on the test data.
>
> Mean and variance of all $|\Delta(u,v)|$ are listed as:
> | Experiment | Mean| Variance|
> |-|-|-|
> |1| 0.19250| 0.00185|
> |2| 0.23114| 0.00497|
>
> Experiment 1 shows that learners with similar training data experiences tend to have similar performance on future practice, supporting our collaborative modeling motivation from the perspective of dataset statistic. Experiment 2 proves the effectiveness and interpretability of learner embeddings learned by Coral. Both experiments reinforce that similarity of learner embeddings by Coral can align with statistical data. We will make this insight clearer in the revised paper.
> > **Weakness 3:** Computational inefficiency.
>
> Our paper provides 3 alternative efficient solutions in Appendix F.2 to improve Coral's computational efficiency. Although these solutions may not guarantee theoretical optimality, the experimental results demonstrate their effectiveness, which prove Coral can be used to some large-scale scenarios. For real-time response, co-disentangled cognitive state (line 249) can be precomputed and stored, with only retrieval needed during online computation. We will continue to optimize the model efficiency in future studies.
> > **Weakness 4:** Concerns on hyperparameter tuning complexity.
>
> Coral requires minimal hyperparameter tuning. We adjust only two parameters, $\beta$ and $K$, with other parameters fixed based on prior experience. Experimental results in Fig3 (c) show $K$ stabilizes after reaching a threshold due to attention control. Thus, when tuning, we can start with a larger $K$ and gradually decrease it. For $\beta$, we recommend values of 0.25, 0.5, and 0.75. Typically, 3 to 4 iterations are sufficient to find satisfactory parameters.
> > **Weakness 5:** Add more baselines.
>
> We conduct additional experiments with four new baselines. **The experimental setups, results and analysis are given in Public Response** (https://openreview.net/forum?id=JxlQ2pbyzS&noteId=XpCCzTJX66).
>
> ---
> [1] Albert Bandura and Richard H Walters. Social learning theory. 1977.
>
> [2] Improving knowledge tracing with collaborative information. WSDM'22.
>
> [3] Homogeneous Cohort-Aware Group Cognitive Diagnosis: A Multi-grained Modeling Perspective. CIKM'23.
>
> [4] Disentangling cognitive diagnosis with limited exercise labels. NeurIPS'24.

---

> ### Author Response · Authors · 2024-08-12
>
> Dear Reviewer Ty5J,
>
> We appreciate your careful reading and detailed feedback of our paper, and giving us the opportunity to answer your questions.
>
> During the rebuttal phase, we have carefully addressed each of your concerns and are eager to receive your feedback for further discussion. We understand and appreciate the effort you put into reviewing the paper, so we have summarized our response for your quick reading:
>
> 1. Our motivation is well-supported by psychological theories, existing literature, and data analysis. Our model is the first exploration in the disentangled collaborative CD direction and is innovative. Please refer to the rebuttal for details.
> 2. There is a misunderstanding concerning our interpretability experiment, which is thoroughly addressed in the case study in Appendix E. Additionally, we further provide dataset-based conclusions in the rebuttal that answer your questions.
> 3. Thank you for your concerns regarding the computational efficiency limitation. We agree that this is a main bottleneck for our Coral model. To address this, our paper presents three preliminary strategies to improve efficiency, as detailed in Appendix F.2, and will focus on improving computational efficiency of Coral in future research. Additionally, we proactively acknowledge the computational efficiency limitations in the limitations section. According to the NeurIPS 2024 checklist guidelines, this proactive acknowledgment is considered a positive practice and should not be penalized.
> 4. Our experiments mainly require the tuning of two hyper-parameters ($\beta$ and $K$) with a fixed learning rate, which is not complex. The hyper-parameter tuning complexity is similar to DCD and RCD in baseline.
> 5. We add four additional baselines in the rebuttal.
>
> Thank you again for your time and effort, and we will revise the paper following each of your suggestions. If you have any further questions or need clarification on any point in our response, please let us know. We will respond promptly.
>
> Best,
>
> Authors.

---

> > ### Comment · Reviewer_Ty5J · 2024-08-13
> >
> > Thank you for the detailed responses and extra efforts. I have increased my score.

---

> ### Author Response · Authors · 2024-08-14
>
> Dear reviewers,
>
> Thank you for recognizing our work as well as our rebuttal. We are pleased to have addressed your concerns and will revise the paper according to your suggestions.
>
> We are very grateful to you for your suggestions on interpretability experiments from the data statistic perspective to help us improve the quality of our papers!
>
> Thank you!
>
> Best,
>
> Authors

---

### Official Review · Reviewer_k7ko · 2024-07-12

**Soundness:** 3
**Presentation:** 3
**Contribution:** 3
**Rating:** 5
**Confidence:** 4

**Summary:**

This paper presents a Collaborative Cognitive Diagnosis model, named Coral, which incorporates a disentangled state encoder, collaborative graphs, and a state decoder. The goal of Coral is to model collaborative and disentangled cognitive states using representation learning techniques.

**Strengths:**

1. The introduction of collaborative connections to cognitive diagnosis is well-motivated and logical.

2. The paper is well-structured, and the methodology is explained in detail.

3. The release of the code enhances the reproducibility of the research.

**Weaknesses:**

1. While the authors identify the challenge of obtaining explicit collaborative connections, they do not provide clear solutions to address this issue.

2. The heavy use of theoretical notations makes the paper less readable.

3. The detailed presentation of the methodology makes the experimental section appear crowded.

4. The performance improvements demonstrated by Coral are moderate.

**Questions:**

1. What is the performance impact if the concepts are not disentangled? Currently, there is insufficient motivation for integrating collaborative signals into the disentangled representations.

**Limitations:**

There is no significant negative societal impact identified in this work. The authors acknowledge an inefficiency issue within the model.

---

> ### Author Rebuttal · Authors · 2024-08-05
>
> We appreciate your careful reading and detailed discussion of our paper. We address your concerns below and please let us know if there are remaining questions or unclear points.
>
> For the weaknesses:
> ---
> > **Weakness 1:** Explain the solution of collaborative relation construction in detail.
>
> Thank you for your feedback and for giving us the opportunity to clarify our solution.
>
> Constructing collaborative connections aims to find $K$ collaborative neighbors for each learner $u \in V$ and build the collaborative graph $G$ (see lines 11-31 in Algorithm 1). This process is formulated as $ \max P(G|V,Z) $, where $V$ is the set of learners and $Z$ is the learner state vector. To achieve this, we iteratively find the $K$ optimal neighbors for each learner $u$ with the following initialization: $u$'s neighbor set initially includes only $u$, i.e., $ r_{u}^{0} = {u} $, and the non-neighbor set consists of all other learners, i.e., $ V_u = V \setminus {u} $.
> The iteration starts and repeats $K$ times until $u$ has $K$ neighbors. Each iteration includes the following steps:
>
> **Iteration (for $ k = 1, 2, \ldots, K $):**
>    - Compute context vector of $u$ using $(k-1)$ neighbors: $ \text{Context}(u) = \sum_{v \in r_{u}^{k-1}} Z_v $
>    - Calculate similarity scores between $u$'s context and each non-neighbor learner $v$: $ \text{score}_{u,v} $ for each $ v \in V_u $
>    - Select the most similar learner as the $k$-th neighbor: $b_{u}^k = \arg\max_{v \in V_u} \text{score}_{u,v} $
>    - Update neighbor set: $ r_{u}^{k} = r_{u}^{k-1} \cup \{b_{u}^k\} $
>    - Update non-neighbor set: $ V_u = V_u \setminus \{b_{u}^k\} $
>
> We sincerely hope our explanation resolves your concerns. We will include more examples in the revised version for clarity.
>
> > **Weakness 2 and  Weakness 3:** Issue of balance between theoretical details and experiments.
>
> We apologize for any confusion. To enhance readability, we included detailed proofs and algorithms in the appendix while summarizing key conclusions in the main text of the paper. For instance, the core construction method of the collaborative graph is summarized in lines 202-207 of the main text, with detailed derivations in Appendix B and algorithms in Appendix C. This structure ensures that our key conclusions are understandable without requiring a deep dive into the theoretical derivations, as acknowledged by reviewers vXdf, t9qg, bUz3, and UPTU.
>
> We will simplify the theoretical symbols in the revised version and include more examples to improve clarity. Additionally, we will increase the size and spacing of Fig.3 and 4 in experimental section.
>
> > **Weakness 4:** Concerns on performance improvement.
>
> Coral's performance improvements are validated through experiments across three different scenarios (line 296), **not solely in Table 1**. We sincerely encourage you to consider the model's performance across all scenarios to appreciate Coral's comprehensive enhancement.
>
> Specifically, Table 1 lists results for the normal scenario (20% test data), where Coral demonstrates significant improvements on Junyi and ASSIST. The improvement on the NIPS202EC dataset is relatively modest, primarily because the NIPS202EC dataset is very dense (as shown in Table 3, with 367.5 problem-solving records per learner), leading to smaller differences between models. Additionally, Figures 2(a) and (b) illustrate significant performance improvements in scenarios with data sparsity and cold start, which are commonly encountered in real platforms, showing that Coral significantly outperforms the baselines.
>
> For the questions:
> ---
> > **Question 1:** Concept disentanglement impacts and insufficient motivation for collaboration modeling.
>
> - **Concept disentanglement impacts:**
> To answer your first question, we conduct experiments removing the disentanglement on concepts by representing each learner with a single $C$-dimensional vector instead of the original $C$ $d$-dimensional vectors, denoted as w/o VecDis. The following experimental results on NIPS2020EC prove the effectiveness of disentanglement.
> |Model|ACC|AUC|F1|RMSE|
> |-|-|-|-|-|
> |w/o VecDis|0.71523|0.78331|0.72667|0.43412|
> |**Coral**|**0.71622**|**0.79103**|**0.72890**|**0.43200**|
> - **Clarify the motivation:**
>
> Thank you for your attention to our motivation and for giving us the opportunity to clarify. In fact, our core motivation is to integrate collaborative signals into CD. Based on this motivation, disentangled representations are introduced to manage the increased complexity of modeling both individual and collaborative representations. Detailed explanations are as follows:
>
> 1. Motivation 1 (collaborative Modeling): Introducing collaborative signals between learners in CD models enhances both performance and interpretability. It is sufficiently supported by social learning theories [1], data analysis (see experiments 1 and 2 in the response to Weakness 2 of Reviewer Ty5J), and empirical effectiveness on learner sequences [2] and group modeling [3], demonstrating the benefits of incorporating collaborative signals.
> 2. Motivation 2 (disentanglement): To model both individual and collaborative learner representations following motivation 1, complexity increases. Traditional single-vector representations may not adequately capture this complexity. Thus, disentanglement is naturally introduced to convert single vector into multiple vectors, reducing dimensional correlation and enhancing modeling capability, inspired by [4].
>
> Thank you again, and we sincerely hope our explanation resolves your concerns. We will make this insight clearer in a revised paper.
>
> ---
> [1] Albert Bandura and Richard H Walters. Social learning theory. 1977.
>
> [2] Improving knowledge tracing with collaborative information. WSDM'22.
>
> [3] Homogeneous Cohort-Aware Group Cognitive Diagnosis: A Multi-grained Modeling Perspective. CIKM'23.
>
> [4] Disentangling cognitive diagnosis with limited exercise labels. NeurIPS'24.

---

### Official Review · Reviewer_UPTU · 2024-07-12

**Soundness:** 3
**Presentation:** 3
**Contribution:** 3
**Rating:** 7
**Confidence:** 4

**Summary:**

This paper explores the Coral model, which combines disentangled representation learning with collaborative signals to enhance cognitive diagnosis in intelligent education. It focuses on identifying implicit collaborative links among learners to improve understanding of their cognitive states. The model integrates a context-aware collaborative graph learning mechanism to simulate the collaborative relationships between learners based on their cognitive states. By incorporating collaborative information into the disentangled cognitive states of learners, the model achieves collaborative disentanglement. Experiments on real datasets demonstrate Coral's performance compared to existing methods.

**Strengths:**

1. This paper will undoubtedly attract attention and excitement from researchers in various fields, e.g., personalized learning, cognitive diagnosis, knowledge tracing, and performance prediction. It represents an exploration of modeling user collaboration in educational settings. While the modeling of collaborative relationships among users is well-established in other fields (e.g., social networks, recommendations), the technological development in online education has lagged. The research line on modeling learner relationships is almost non-existent. I hope this work will advance the intelligent education community.
2. Compared to previous cognitive diagnosis research, Coral introduces a new method that combines disentangled representation learning with collaborative signals, holding great potential for enhancing cognitive diagnosis.
3. Theoretical Contribution: The authors have meticulously derived the optimization form of the lower bound of cognitive representation encoders, the approximate optimization objective of collaborative relationship modeling, and the model convergence. The theoretical proofs are reasonable and solid.
4. The experimental results on real datasets in normal, data-sparse, and cold-start scenarios indicate that Coral outperforms existing methods, demonstrating its effectiveness in improving diagnostic accuracy.
5. The writing is good. The proposed problem and model are described clearly.

**Weaknesses:**

I expect that this paper will comfortably clear the bar for acceptance. However, I believe two main issues should be addressed first. Due to these reasons, I have set my score at a relatively borderline level but anticipate increasing it after the authors' rebuttal.

1. If two students have fewer interaction records, leading to increased similarity in their interaction patterns, they might exhibit different cognitive levels. Using the method outlined in the paper to learn their representations may yield similar results, but this might not accurately reflect the true situation.
2. The paper employs GNN to learn network representations, but the graph-based baseline comparison only includes RCD and does not include some recent graph-based cognitive diagnosis methods, such as [1].

Other/Minor issues (Optional):

1. In the second paragraph, the authors describe an example: experienced teachers can summarize students' similarities. Similarly, could large language models be used instead of real human teachers to construct collaborative graphs? I look forward to the authors discussing the integration with LLMs. Although this does not affect the readability and acceptance of the paper, I believe readers will be very interested.
2. In Table 6, "w/ full-kit (m,m)" should be "w/ full-kit (n,m)."
3. In Table 1, why are there two results for DCD on the NIPS2020EC dataset? It seems like results from two different runs.

```
[1] Wang, Shanshan, et al. "Self-supervised graph learning for long-tailed cognitive diagnosis." Proceedings of the AAAI Conference on Artificial Intelligence. Vol. 37. No. 1. 2023.
```

**Questions:**

1. The motivation of this paper is to consider collaborative information among learners. What are the specific differences between the collaborative information proposed in this paper and the collaborative information in recommendation systems?
2. Do the ASSIST and NIPS2020EC datasets have similarities and prerequisite relationships among the knowledge concepts that support RCD?
3. In the example of question recommendation in Appendix E, are these five questions answered by learners in comparable situations (i.e., without acquiring new knowledge during the answering process)?

**Limitations:**

The authors discuss the limitations of the computational efficiency of the paper. Meanwhile, they also give 2 simple yet effective solutions to address current limitations in the appendix.

---

> ### Author Rebuttal · Authors · 2024-08-05
>
> Thank you for the detailed and constructive feedback! We answer your comments and questions below. Please let us know if you have additional questions.
>
> For the weaknesses:
> ---
> > **Weakness 1:** How to mitigate bias issues in establishing learner collaboration relationships with insufficient interaction data.
>
> Thank you for raising this issue. Our variational encoder can mitigate this problem by magnifying subtle differences in the latent feature space. In fact, existing CD models that rely solely on student interaction data typically face this challenge[1].
>
> > **Weakness 2:** Add a graph-based baseline.
>
> Thank you for pointing out the absence of a graph-based baseline, i.e., SCD. **We have conducted learner performance prediction tasks using SCD on three datasets, as shown in our Public Response** (https://openreview.net/forum?id=JxlQ2pbyzS&noteId=XpCCzTJX66).
> Experimental results prove the consistent superiority of Coral compared with SCD. We will report the complete results in the revised version.
>
> For the concerns:
> ---
> > **Concern 1:** How to use LLMs for constructing the collaborative graph?
>
> Thank you for your forward-thinking perspective. We believe that LLMs have significant potential in constructing educational collaboration graphs by designing agents that can identify collaborative learners based on educational prior knowledge and psychological assumptions. Existing studies [2,3,4] have demonstrated the use of LLMs to predict and analyze graphs in text, which can provide valuable insights for constructing collaboration graphs among learners in educational settings. We consider this a valuable research direction and plan to explore it in future studies.
>
> > **Concern 2:** Some typos.
>
> Thank you for pointing out this issue. We will correct the symbols in the revised paper.
>
> > **Concern 3:** Confusion on baselines with duplicate names.
>
> We apologize for the confusion. The first line of the DCD results does not include the tree structure, aligning with the settings for the ASSIST and Junyi datasets, which do not provide tree structures (as described in the "Settings" section). The second line of DCD results includes the tree structure. We will correct this in the revised version and label the second set as DCD_tree.
>
> For the questions:
> ---
> > **Question 1:** Explain the difference of "collaboration" between education and recommendation.
>
> The core idea in both contexts is to model similarities between entities to aid predictions. In the educational scenario, "collaboration" considers cognitive similarities between learners, aiming to diagnose the learner's cognitive state rather than predict the final outcome of answering questions. In a recommendation system, "collaboration" models the similarity of preferences among users, aiming to predict similar users' decisions on the same item through similarity.
>
> > **Question 2:** How to construct conceptual relations from ASSIST and NIPS2020EC datasets for running RCD?
>
> The two datasets do not provide the concept maps required by RCD. Instead, we constructed the conceptual relations from exercise data using the statistical tool (https://github.com/bigdata-ustc/RCD) proposed by RCD.
>
> > **Question 3:** Details about the case study.
>
> The example questions in the case study are randomly selected without a specific recommendation strategy, aiming to showcase the model's performance and interpretability in cold-start scenarios. Regarding "comparable situations", note that during inference, all model parameters remain fixed, meaning learners do not gain new knowledge. We will provide more details in the revised version.
>
> ---
>
> [1] Towards the Identifiability and Explainability for Personalized Learner Modeling: An Inductive Paradigm[C]//Proceedings of the ACM on Web Conference 2024. 2024: 3420-3431.
>
> [2]  Leveraging Large Language Models for Concept Graph Recovery and Question Answering in NLP Education[J]. arXiv preprint arXiv:2402.14293, 2024.
>
> [3] Large Language Model (LLM)-enabled Graphs in Dynamic Networking[J]. arXiv preprint arXiv:2407.20840, 2024.
>
> [4] Canonicalize: An LLM-based Framework for Knowledge Graph Construction[J]. arXiv preprint arXiv:2404.03868, 2024.

---

> > ### Comment · Reviewer_UPTU · 2024-08-08
> >
> > I appreciate the authors’ response and the additional experiments. I have revised my review score to Accept.

---

> ### Author Response · Authors · 2024-08-08
> **Official Comment by Authors**
>
> Thank you for your prompt feedback and for recognizing the value of our work!
>
> We are pleased to have addressed your concerns and will revise the paper according to your suggestions. If you have any further questions, please feel free to bring them up for discussion. We will address your concerns as quickly as possible!
>
> Once again, thank you for your support and constructive feedback.

---

### Official Review · Reviewer_bUz3 · 2024-07-12

**Soundness:** 3
**Presentation:** 3
**Contribution:** 3
**Rating:** 7
**Confidence:** 4

**Summary:**

The authors propose Coral, a Collaborative Cognitive Diagnosis model with Disentangled Representation Learning, aimed at improving our understanding of human learning by leveraging collaborative signals among learners. By disentangling cognitive states and dynamically constructing a collaborative graph, Coral captures implicit connections between learners, leading to more accurate cognitive diagnoses. The model's innovative approach involves an iterative process of encoding, collaborative representation learning, and decoding, which significantly enhances its performance over existing methods. Extensive experiments demonstrate Coral's superior effectiveness in real-world datasets, making it a promising tool for intelligent education systems.

**Strengths:**

1. Coral’s use of disentangled representation learning and dynamic graph construction is a novel approach in the field. This allows the model to capture implicit collaborative signals among learners, leading to more accurate and insightful cognitive diagnoses.

2. The extensive experiments and superior performance across multiple real-world datasets highlight Coral’s practical utility.
By disentangling cognitive states and integrating collaborative information, Coral offers greater explainability and controllability in cognitive diagnosis. This may make it easier for educators and researchers to understand and interpret the learning patterns and cognitive states of students.

**Weaknesses:**

1. Coral's effectiveness relies heavily on the quality and quantity of learner data available. In scenarios where such data is sparse or noisy, the model's performance may not be as robust.

2. While the model shows great promise, its computational requirements for dynamically constructing collaborative graphs and learning representations might pose scalability issues, especially in very large educational environments.

3. There may lack some highly related baselines/papers for the cognitive diagnosis.

[1] https://arxiv.org/pdf/2002.00276
[2] https://ieeexplore.ieee.org/document/10027634
[3] https://arxiv.org/pdf/2005.13107

**Questions:**

See weaknesses.

---

> ### Author Rebuttal · Authors · 2024-08-05
>
> Thank you for the detailed and constructive feedback! We answer your comments and questions below. Please let us know if you have additional questions.
>
> > **Weakness 1:** Robustness under data sparsity or noise scenarios.
>
> Thank you for highlighting the robustness of CD in data sparsity or noise scenarios.
>
> **In fact, we have presented experiments under data sparsity and cold start scenarios in Fig3(a) and (b),** which demonstrate that Coral exhibits better robustness compared to the baseline models.
>
> **Additional experiments with noisy data:**
> Due to the lack of noisy labels, current CD models often struggle with data noise issue. Educational data noise is typically caused by implicit learner errors or guesses, which are difficult to quantify. Therefore, existing methods often treat noise as latent trainable parameters (e.g., [1][2]), but they fail to validate the model's ability to handle noise since the extent of noise in the dataset is unknown. To address your concerns about noisy data, we explicitly simulate noisy training scenarios by randomly flipping the performance labels of 5% of learners in each training dataset (0 -> 1, 1 -> 0). The test data remains unchanged and is considered noise-free, allowing us to assess whether models trained on noisy data maintain robustness on a noise-free dataset. Below are some of the results:
>
> |ASSIST with 5% Noise|ACC|AUC|RMSE|junyi with 5% Noise|ACC|AUC|RMSE|NIPS2020EC with 5% Noise|ACC|AUC|RMSE|
> |-|-|-|-|-|-|-|-|-|-|-|-|
> |IRT|0.69577|0.70034|0.45321|IRT|0.79632|0.79314|0.37674|IRT|0.69221|0.74732|0.45621|
> |MIRT|0.71233|0.73674|0.43551|MIRT|0.79314|0.79667|0.38117|MIRT|0.70443|0.71074|0.44152|
> |PMF|0.70544|0.74327|0.45217|PMF|0.78512|0.78617|0.38211|PMF|0.703774|0.77662|0.44513|
> |NCDM|0.70522|0.73316|0.44021|NCDM|0.78334|0.77513|0.40207|NCDM|0.70217|0.77441|0.45122|
> |KaNCD|0.7127|0.75672|0.43641|KaNCD|0.80332|0.80017|0.37724|KaNCD|0.70107|0.77347|0.44102|
> |RCD|0.70032|0.73147|0.44263|RCD|0.81163|0.81522|0.37233|RCD|0.70732|**0.78533**|0.44324|
> |**Our Coral**|**0.72174**|**0.75807**|**0.43217**|**Our Coral**|**0.81873**|**0.81751**|**0.37027**|**Our Coral**|**0.70965**|**0.78533**|**0.44132**|
>
> These results clearly show that Coral exhibits greater robustness compared to baselines in noisy scenarios.
> We will add all experimental results and discuss the impact of noisy data in the limitations section of the revised paper.
>
> > **Weakness 2:** Limited computational efficiency.
>
> Thank you for your feedback on computational efficiency.
>
> Our paper has presented three effective solutions to improve the efficiency of Coral in Appendix F.2. These strategies significantly enhance Coral's computational efficiency. While they may not guarantee theoretical optimality, the experimental results demonstrate substantial improvements, which prove Coral can be used to some large-scale scenarios. For real-time response, co-disentangled cognitive state (line 249) can be precomputed and stored, with only retrieval needed during online computation. We will continue to optimize the model efficiency in future studies.
>
> > **Weakness 3:** Add three related baselines.
>
> Thank you for pointing out this issue and giving us the opportunity to supplement baselines.
>
> We conduct experiments for the three baselines you suggested on three datasets. **The experimental setups, results and analysis are available in the Public Response** (https://openreview.net/forum?id=JxlQ2pbyzS&noteId=XpCCzTJX66).
> Experimental results demonstrate that Coral consistently outperforms these baselines across all datasets.
> We will add the complete experimental results and introduce these significant related works in the revised version.
>
> ---
>
> We hope these responses comprehensively address your points and enhance your understanding of our work. We greatly appreciate your constructive feedback and remain open to any further inquiries or suggestions for improvement. Thank you for your support and valuable contribution to improving our paper.
>
> ---
>
> [1] Terry A Ackerman. Multidimensional item response theory models. Wiley StatsRef: Statistics Reference Online, 2014.
>
> [2] https://ieeexplore.ieee.org/document/10027634

---

> > ### Comment · Reviewer_bUz3 · 2024-08-08
> >
> > Your additional experiments under noisy data scenarios are valuable. The explicit simulation of noisy training conditions and the clear presentation of results demonstrate Coral's robustness effectively.
> >
> > While it might be beneficial to also discuss potential future work on handling different types of noise, such as systematic biases or varying noise levels, we understand that this may be beyond the scope of the current study focused on cognitive tasks (does not require additional designed experiments).
> >
> > Addtionally, the inclusion of experiments with the suggested baselines is a significant improvement.
> >
> > Once again, thank you for your hard work and the thoughtful rebuttal. Your contributions are valuable, and I would like to improve my score.

---

> ### Author Response · Authors · 2024-08-08
>
> Thank you for your prompt feedback and for recognizing the value of our work!
>
> We are pleased to have addressed your concerns and will revise the paper according to your suggestions. If you have any further questions, please feel free to bring them up for discussion. We will address your concerns as quickly as possible!
>
> Once again, thank you for your support and constructive feedback.

---

### Official Review · Reviewer_t9qg · 2024-07-28

**Soundness:** 3
**Presentation:** 3
**Contribution:** 4
**Rating:** 6
**Confidence:** 3

**Summary:**

This paper introduces Coral, a novel model for collaborative cognitive diagnosis with disentangled representation learning, that can help in developing intelligent education systems. The key contributions are incorporating collaborative information among learners to improve cognitive state diagnosis by using disentangled representation learning to model cognitive states, and then using context-aware graph construction method to identify implicit collaborative connections. Coral demonstrates superior performance over state-of-the-art methods across several real-world datasets, showing particular strength in sparse and cold-start scenarios. The approach offers improved interpretability and adaptability in modeling learners' cognitive states, potentially advancing personalized tutoring and understanding of human learning processes.

**Strengths:**

* The paper introduces an intriguing dual-perspective approach to cognitive diagnosis. By examining both inner-learner and inter-learner aspects, it offers a more comprehensive view of cognitive states. The integration of context-aware collaborative graph learning with disentangled representation learning is particularly noteworthy, as it potentially allows for more nuanced insights into learner interactions.

* The empirical validation using three diverse datasets (ASSIST, Junyi, NIPS2020EC) lends credibility to the model's effectiveness. Its superior performance across various metrics compared to existing models is encouraging. Moreover, the model's resilience in scenarios with sparse data and cold start environments, suggests practical applicability in real-world educational settings.

* From a structural standpoint, the paper presents its ideas coherently. The authors have taken care to provide detailed explanations and mathematical formulations for each component of their model. The inclusion of key equations and visual aids, such as t-SNE plots, effectively illustrates the model's inner workings, particularly in terms of disentanglement and neighbor selection processes.

* This research makes a valuable contribution to the field of cognitive diagnosis. By enhancing both the accuracy and interpretability of learner cognitive state assessments, it opens up new avenues for personalized education. The model's broad applicability across various educational datasets, coupled with its novel methodologies like context-aware graph learning and co-disentanglement, provides a solid foundation for future research in educational data mining and related fields.

**Weaknesses:**

* The model suffers from computational inefficiencies, which can limit its practical applicability, especially in large-scale educational settings. This issue is transparently acknowledged by the authors too, which is positive.

* While the paper aims to achieve disentangled representations, the evaluation of disentanglement quality remains simple and limited. It will be valuable to add more ablation studies related to disentanglement quality and run qualitative analysis of how individual latent dimensions correspond to interpretable knowledge concepts across different datasets.

* The paper doesn't adequately address potential limitations of the collaborative approach, such as negative transfer or performance in edge cases. It will be valuable to discuss scenarios where the collaborative approach might not be beneficial or could lead to negative transfer. E.g., a highly advanced learner who struggles in one specific area may be grouped with generally low-performing learners

**Questions:**

* How sensitive is the model performance to changes in key hyperparameters such as the number of neighbors (K) and the beta value? Adding a sensitivity analysis for these hyperparameters in the appendix can help assess the model's robustness and guide practitioners in tuning it for different datasets.

* Can you provide a breakdown of Coral's performance as the percentage of questions answered by new learners varies (e.g., 0%, 5%, 10%, 20%)? This would give a clearer picture of the model's efficacy in true cold-start scenarios.

* Is it possible to show how disentangled factors correspond to specific knowledge areas (since mapping to interpretable cognitive factors are not easy) across different datasets? This would strengthen the claim of improved interpretability over existing methods.

**Limitations:**

* The authors briefly mention computational inefficiencies as a limitation in Section 5. They also provide some preliminary optimization strategies in Appendix F.2, which is positive. However, it would be valuable to also expand it to other concerns like any potential bias in the modeling process, limitations in the datasets used or generalizability to other educational contexts, any limitations in the disentanglement approach to capture all relevant cognitive factors, etc

* Also the authors do not fully address potential ethical concerns and mitigation strategies related to privacy and fairness in cognitive modeling. E.g., privacy concerns related to modeling learner cognitive states, potential bias in the model that could disadvantage certain groups of learners, etc

---

> ### Author Rebuttal · Authors · 2024-08-05
>
> Thank you for the positive remarks and insightful feedback! We address your comments and questions below. Please let us know if you have additional questions.
>
> > **Weakness 1:** Concerns on computational efficiency.
>
> Thank you for acknowledging our discussion of the model's efficiency limitation.
>
> In fact, we also provide alternative efficient solutions in Appendix F.2 to address this limitation, where we design three practical strategies to improve Coral's computational efficiency. Although these solutions may not guarantee theoretical optimality, the experimental results demonstrate their effectiveness.
>
> > **Weakness 2:** Disentanglement evaluations.
>
> Thank you for your suggestion on evaluating disentanglement quality.
>
> Our disentanglement involves decoupling each learner into $C$ vectors (as vector-level disentanglement) and using KL divergence at the element level to encourage independence between each dimension of each vector based on Eq.5 (as dimension-level disentanglement). The current paper assesses dimension-level disentanglement quality by: (1) quantitative correlations between dimension independence level (IL) and prediction performance (Fig.3(a)), (2) qualitative embedding visualizations (Fig.4(b)), and (3) an ablation study of dimension-level disentanglement by discarding the KL term ("w/o KL" in Tab.2).
>
> To further address your concerns, we add two additional studies:
>
> 1. We set different values for $\beta$ to explore the effect of varying disentanglement strengths (KL term in Eq. 5) on IL (i.e., dimensions' independence level of disentangled vectors) and performance, inspired by [1]. The experimental results on ASSIST are as follows:
>
> |Value of $\beta$|0.25|0.5|0.75|1|
> |-|-|-|-|-|
> |ACC|0.71164|0.72533|0.72130|0.72017|
> |AUC|0.74318|0.77312|0.77151|0.76817|
> |IL|0.82173|0.85031|0.87263|0.87342|
>
> The results indicate that as the disentanglement strength increases, the IL metric gradually increases, demonstrating that each dimension of learner representations becomes more independent. The model's predictive performance initially improves and then slightly decreases after reaching a threshold ($\beta = 0.5$). This suggests that both too low and too high disentanglement strengths can affect the model's performance.
>
> 2. We conduct ablation studies to remove the vector-level disentanglement by representing each learner with a single $C$-dimensional vector instead of the original $C$ d-dimensional vectors, denoted as "w/o VecDis". The experimental results on NIPS2020EC are as follows:
> |Model|ACC|AUC|RMSE|
> |-|-|-|-|
> |w/o KL|0.71026|0.78990|0.44382|
> |w/o VecDis|0.71523|0.78331|0.43412|
> |**Coral**|**0.71622**|**0.79103**|**0.43200**|
>
> These results demonstrate the effectiveness of vector-level disentanglement. We will report the full results in a revised paper.
>
> > **Weakness 3:** How to address potential negative transfer impact in the collaborative graph.
>
> Our model addresses potential negative transfer through attention weights (see Eq.8 for $s_{u,v}$), as discussed on lines 230-233. When computing attention between learners $u$ and $v$, both cognitive similarity and contextual similarity are considered. This approach ensures that attention is low for non-similar and non-collaborative neighbors, thus mitigating negative transfer impacts.
>
> We conduct additional ablation experiments to further support this clarity:
>
> |ASSIST|ACC|AUC|RMSE|junyi|ACC|AUC|RMSE|NIPS2020EC|ACC|AUC|RMSE|
> |-|-|-|-|-|-|-|-|-|-|-|-|
> |w/o attention|0.71996|0.75023|0.42533|w/o attention|0.82172|0.82969|0.37112|w/o attention|0.71173|0.78334|0.43430|
> |**Coral**|**0.72533**|**0.77312**|**0.42034**|**Coral**|**0.82534**|**0.83503**|**0.36403**|**Coral**|**0.71622**|**0.79103**|**0.43200**|
>
> The model without attention (w/o attention) performs worse compared to Coral, demonstrating the effectiveness of attention in managing negative transfer. We will include the full results in the revised paper and emphasize this technical advantage.
>
> > **Question 1:** Concerns on hyper-parameter sensitivity experiments.
>
> Thank you for your feedback.
>
> Current paper already included hyper-parameter experimental results (Fig.3) and findings  (lines 323-328) on the number of neighbors $K$.
>
> Additionally, **we conduct experiments on different embedding dimensions $d$, as detailed in the Public Response** (https://openreview.net/forum?id=JxlQ2pbyzS&noteId=4eDEmw2q38).
>
>
> > **Question 2:** Data sparsity impacts.
>
> Current paper provides experimental results of Coral with 20%, 40%, 60%, and 80% sparsity (Fig.3(a)). To fully address your concerns, we further conduct experiments with 5% and 10% sparsity as follows:
> |ACC|IRT|MIRT|NCDM|KaNCD|RCD|**Coral**|
> |-|-|-|-|-|-|-|
> |5%|0.57073|0.59274|0.609715|0.60332|0.64113|**0.64933**|
> |10%|0.64231|0.66774|0.67752|0.67321|0.66432|**0.67933**|
>
> The experiments show the robustness of Coral. For 0% sparsity, current CD research that rely on practice data cannot effectively handle this scenario due to the need for initial data from each learner.
> > **Question 3:** Interpretability.
>
> Each learner is disentangled into $C$ vectors, so that the disentanglement representation can naturally correspond to a specific knowledge, e.g., $z_{u}^{(c)}$ is $u$'s cognitive state of knowledge $c$.
>
> An additional interpretability case study is given in Appendix E.
>
> > **Limitations:**
>
> Thank you for your feedback. We supplement with a PISA 2015 data to expand dataset diversity (**see the Public Response**). To address data privacy, federated learning or cross-domain modeling can be integrated to enhance Coral's privacy-preserving capabilities. For fairness, we will ensure equitable outputs by identifying and mitigating the impact of sensitive attributes on model predictions. In the future, we will also design more powerful disentanglement algorithms.
>
> We will include a discussion on these technical limitations and ethical considerations.
>
> ---
> [1] Disentangling cognitive diagnosis with limited exercise labels. NeurIPS'24.

---

> > ### Comment · Reviewer_t9qg · 2024-08-10
> >
> > For question 2, authors discuss the performance of Coral in sparse scenario, whereas my question was around performance in cold start scenario. This would involve testing the model with new learners who have answered 5%, 10%, 20% of questions and comparing the results to understand how well the model handles true cold-start scenarios
> >
> > Saying that, I would like to thank the authors for performing additional experiments and providing other clarifications. After reviewing the results, I would like to maintain my previous scores

---

> ### Author Response · Authors · 2024-08-10
> **Further Response to Question 2**
>
> Thank you for recognizing the value of our work! We are pleased to have addressed most of your concerns and will incorporate the suggested modifications into the revised version.
>
> We apologize for the misunderstanding regarding Question 2. Your further explanation helped us better understand this concern.
> Regarding **Question 2**, we would like to first clarify that such 5% and 10% cold-start phenomena are rare in real-world platforms unless it is a completely new platform. Current CD research focuses on learner-side cold-start scenarios where:
> 1. The learning platform has already accumulated rich learner data, and only a portion of learners are new [1].
> 2. A new business domain lacks learner data, but there is existing learner data from other related businesses [2].
>
> In fact, **our paper's Figure 3(b) has already validated the model performance under the second cold-start setup**, where the training data is the mature domain and the test scenario is the cold-start domain.
>
> Based on your detailed feedback, we further conduct the suggested **cold-start experiments**: selecting the earliest 5% and 20% of exercise data from learners in chronological order to train the model and then validating performance.
>
> |ACC|IRT|NCDM|KaNCD|**Coral**|
> |-|-|-|-|-|
> |**New** learners with 5% data|0.56932|0.60573|0.60154|**0.64667**|
> |**New** learners with 10% data |0.64228|0.67761|0.67296|**0.67993**|
>
> The results remain consistent with the 5% and 10% sparsity settings. We infer that there are three main reasons for these experimental outcomes:
> 1. In online learning, most learners have very few records. Thus, low sparsity rates like 5% and 10% result in almost no data, with an average of only 3 and 6 records per learner in our ASSIST dataset. This sparsity situation closely resembles that of new learners with minimal data.
> 2. CD assumes a learner’s cognitive state remains constant, meaning the order in which questions are answered does not affect performance [3]. This means that for CD models, there is little difference between using the earliest 5% and 10% of data (cold-start) based on time and using a randomly selected 5% and 10% of data (data sparsity).
> 3. Online education allows learners to practice autonomously, so their early data typically does not suffer from focusing on only a small subset of questions. This enables us to train parameters for most questions even with a limited amount of data.
>
> Thank you for your feedback and for the further explanation, which helped me improve our paper and better understand your concerns. We sincerely hope that our further clarification addresses your concerns.
>
> ---
>
> [1] BETA-CD: A Bayesian meta-learned cognitive diagnosis framework for personalized learning. AAAI'23.
>
> [2] Zero-1-to-3: Domain-Level Zero-Shot Cognitive Diagnosis via One Batch of Early-Bird Students towards Three Diagnostic Objectives. AAAI'24.
>
> [3] Towards a New Generation of Cognitive Diagnosis. IJCAI'21.

---

### Official Review · Reviewer_vXdf · 2024-07-30

**Soundness:** 3
**Presentation:** 3
**Contribution:** 3
**Rating:** 7
**Confidence:** 4

**Summary:**

The paper presents a novel model called Coral, aimed at enhancing cognitive diagnosis in educational contexts by leveraging collaborative signals among learners. The authors argue that learners with similar cognitive states often exhibit comparable problem-solving performances, and thus, understanding these collaborative connections can significantly improve the diagnosis of individual cognitive states.
Key Contributions -
1. Disentangled Representation Learning: Coral introduces a disentangled state encoder to effectively separate the cognitive states of learners. This approach allows for clearer interpretations of individual knowledge proficiencies by disentangling the various cognitive factors that influence learning performance.
2. Collaborative Graph Construction: The model incorporates a context-aware collaborative representation learning mechanism that dynamically constructs a graph of learners. This graph is built by identifying optimal neighbors based on their cognitive states, facilitating the extraction of collaborative information.
3. Co-disentanglement Process: Coral achieves co-disentanglement by aligning the initial cognitive states with collaborative states through a decoding process. This integration enhances the model's ability to reconstruct practice performance while providing a more nuanced understanding of learners' cognitive states.
4. Empirical Validation: The paper reports extensive experiments demonstrating that Coral outperforms state-of-the-art methods across several real-world datasets, highlighting its effectiveness in accurately diagnosing cognitive states through collaborative learning.
Overall, this work addresses significant gaps in existing cognitive diagnosis methods by simultaneously modeling both individual and collaborative learning information, paving the way for more personalized educational interventions.

**Strengths:**

Originality
The paper introduces a unique approach to cognitive diagnosis by integrating collaborative signals among learners, which has not been extensively explored in existing literature. The concept of using a disentangled representation to model cognitive states while simultaneously capturing collaborative learning dynamics is innovative. Prior models either focused solely on individual cognitive factors or did not adequately account for collaborative interactions. I think it reflects a creative synthesis of ideas from domains of educational psychology and collaborative filtering.

Quality
The authors provide a comprehensive framework that includes a disentangled state encoder, a context-aware collaborative graph learning mechanism, and a co-disentanglement process. The empirical validation through extensive experiments on real-world datasets demonstrates the robustness of the model. The results indicate significant improvements over state-of-the-art methods, showcasing the effectiveness of the proposed approach. The paper also includes detailed mathematical formulations and algorithms, which enhance the reproducibility of the research.

Clarity
The paper is well-structured, with a logical progression from the introduction of the problem to the presentation of the model and its components. The use of clear diagrams and figures aids in the understanding of complex concepts, such as the collaborative graph construction and the disentanglement process. The performance comparison table is extremely effective in showcasing how the contribution is significantly better than the state-of-the-art methods on multiple datasets.

Significance
The work definitely shows potential impact on the field of intelligent education. By improving the accuracy of cognitive diagnosis, the Coral model can facilitate more personalized learning experiences. The integration of collaborative learning dynamics addresses a critical gap in existing models and could influence the design of educational technologies and interventions, making this research valuable not only academically but also practically in educational settings.

**Weaknesses:**

1. Limited Generalizability of Experiments
While the paper demonstrates the effectiveness of the Coral model through extensive experiments on several real-world datasets, the choice of datasets may limit the generalizability of the findings. The experiments primarily focus on specific educational contexts, which may not capture the diverse learning environments and learner behaviors present in broader educational settings.
Actionable Insight: Future work should consider testing the model on a wider variety of datasets, including those from different educational domains (e.g., K-12, higher education, online learning platforms) and varying learner demographics. This would help validate the robustness of the model and its applicability across different contexts.
2. Lack of Comparison with Baseline Models
The paper claims significant improvements over state-of-the-art methods, but it does not provide a comprehensive comparison with a more wider range of baseline models. Especially collaborative filtering methods and other machine learning approaches that could serve as alternatives.
Actionable Insight: The authors should include a more diverse set of baseline models in their experiments to provide a clearer picture of Coral's performance. Additionally, discussing the strengths and weaknesses of these models in comparison to Coral would enhance the reader's understanding of the model's contributions.
3. Insufficient Exploration of Hyper-parameter Sensitivity
The model's performance could be sensitive to various hyper-parameters, such as the number of neighbors in the collaborative graph or the dimensions of the disentangled representations. However, the paper does not provide a thorough analysis of how these hyper-parameters impact the model's performance.
Actionable Insight: Conducting a sensitivity analysis on key hyper-parameters would help identify optimal settings and provide insights into the model's robustness. This could involve systematic experiments varying one parameter at a time while keeping others constant, followed by a discussion of the results.
4. Explainability and Interpretability
While the paper emphasizes the importance of disentangled representations for explainability, it lacks a detailed exploration of how the model's outputs can be interpreted in practical educational contexts.
Actionable Insight: The authors should include case studies or examples demonstrating how the model’s outputs can be interpreted and applied in real educational settings. This could involve visualizing the relationships between cognitive states and learner performance or providing guidelines for educators on how to utilize the model's insights.
5. Potential Overfitting Concerns
The complexity of the Coral model, with its multiple components and parameters, raises concerns about potential overfitting, especially if the training datasets are not sufficiently large or diverse. The paper does not address how the model's complexity is managed or validated against overfitting.
Actionable Insight: Providing validation metrics would strengthen the paper's claims regarding the model's generalization capabilities.
6. Limited Discussion on Ethical Considerations
The paper does not address potential ethical implications of using collaborative cognitive diagnosis models in educational settings, such as data privacy concerns or the impact on learner autonomy and agency.
Actionable Insight: Including a section on ethical considerations would enhance the paper's comprehensiveness and relevance. This could involve discussing how to ensure data privacy, the implications of algorithmic bias, and strategies for maintaining learner agency in personalized educational interventions.

**Questions:**

Dataset Diversity: What criteria did you use to select the datasets for your experiments? How do you ensure that the datasets adequately represent the diversity of learning environments and learner behaviors?

Baseline Comparisons: Can you provide more details on the baseline models you compared Coral against? Specifically, how do these models differ in their approach to cognitive diagnosis, and why were they chosen as benchmarks?

Hyper-parameter Sensitivity: Did you conduct a sensitivity analysis on the hyper-parameters of the Coral model? If so, what were the findings, and how do hyper-parameter choices impact the model's performance?

Interpretability of Results: How can educators and practitioners interpret the outputs of Coral in practical settings? Are there specific examples or case studies that illustrate how the model's insights can be applied in educational contexts?

Overfitting Concerns: What measures did you take to prevent overfitting in your model? Can you provide validation metrics or results from cross-validation to support the generalization capabilities of Coral? In your "data_loader.py" script you have provided a check for validation set on line 60 ("val_set.json") and also in the paper on line 279 under the Experimental setup --> Settings section you claim "split all the datasets with a 7:1:2 ratio into training sets, validation sets, and testing sets". However, no validation metrics or validation sets seem to be used.

Ethical Considerations: Have you considered the ethical implications of using collaborative cognitive diagnosis models in educational settings? How do you address potential concerns regarding data privacy and algorithmic bias?

**Limitations:**

The authors have not addressed the limitations and potential negative societal impact of their work, but there are still areas that could be improved. They could further elaborate on the generalizability of their findings beyond the specific educational contexts covered in the experiments. The authors do not explicitly discuss the potential negative societal impact of their work. This is an important aspect that should be addressed, as cognitive diagnosis models can have significant implications for learners, educators, and educational institutions. An in depth explanation should be provided pertinent to limitations on dimensions like : Data Privacy, Algorithmic Bias, Misuse of Diagnostic Information etc.

---

> ### Author Rebuttal · Authors · 2024-08-05
>
> Thank you for the positive remarks and insightful feedback! We answer your comments and questions below. Please let us know if you have additional questions.
> > **Weakness 1:** Concerns on data diversity.
>
> Thank you for your concerns about the datasets.
>
> We would like to emphasize that our datasets are diverse, covering both K-12 scenarios (ASSIST) and online learning scenarios (Junyi and NIPS2020EC). These datasets are also representative, as they are benchmark datasets in CD research. Nearly all top-tier work (e.g., [1][2][3]) uses these datasets for experiments.
>
> Following your suggestion, we further add a new dataset for experiments. **The experimental results are detailed in the Public Response** (https://openreview.net/forum?id=JxlQ2pbyzS&noteId=XpCCzTJX66), supporting our model's effectiveness across diverse domains.
> > **Weakness 2:** Add baselines.
>
> Thank you for your feedback.
>
> We add four related baselines and compare them across three datasets. **The experiments settings, results and model comparison analysis are listed in Public Response** （https://openreview.net/forum?id=JxlQ2pbyzS&noteId=XpCCzTJX66), demonstrating the superiority of Coral over these baselines. We will add complete results and detailed model comparison analysis in a revised version.
>
> > **Weakness 3 and Question 3:** Concerns on hyper-parameter sensitivity experiments.
>
> Thank you for your concern on hyper-parameter sensitivity experiment.  We would like to emphasize that our paper already includes hyper-parameter experiments (Fig.3(c)) on the number of neighbors $K$.
>
> Additionally, we conduct experiments on different embedding dimensions $d$, with values of 124, 200, 256, 521, and 1024. **The experimental results of Coral on ACC are listed in Public Response**（https://openreview.net/forum?id=JxlQ2pbyzS&noteId=XpCCzTJX66), demonstrating that within an appropriate range, the dimension $d$ does not significantly affect Coral's performance. We will include the complete results in the revised version.
> > **Weakness 4 and Question 4:** Interpretability concerns.
>
> We have already provided an interpretable example in Appendix E. This example demonstrates how Coral reasonably infers each learner's future performance by referencing the cognitive states of similar learners, thereby providing educators with a basis for their inferences.
>
> Additionally, we would like to clarify that Coral's interpretability comes from its ability to use cognitive states of similar learners to predict how learners will perform on unfamiliar problems. Disentanglement is a technique to enhance the modeling of collaborative relations, not the source of interpretability. We infer that you might be misled by "co-disentanglement" in line 77, which means aligning learner representations from two perspectives to optimize collaborative relationship modeling, thereby enhancing interpretability. We will make this insight clearer in a revised paper.
> > **Weakness 5 and Question 5:** Overfitting issue.
>
> Thank you for the insightful feedback.
>
> We use an early stopping strategy to mitigate overfitting, stopping training when model's ACC (validation metric) on validation set stabilizes.
> Each model is trained 5 times with a repartition of data, and the average score is reported, similar to 5-fold cross-validation. We provide Coral's ACC error bars for 5 runnings on ASSIST, Junyi, and NIPS2020EC, $\pm 0.00023$, $\pm 0.00017$, and $\pm 0.00031$, ensuing robustness.
>
> In fact, our paper verified Coral's robustness in both sparse and diverse scenarios:
> Fig. 3(a) shows experiments on data sparsity, showing Coral's robustness in handling overfitting with insufficient data. Fig. 3(b) shows cold start experiments where new knowledge is introduced to the test data, increasing test scenario diversity, proving the model's robustness even when the training data lacks diversity.
>
> > **Weakness 6 and Question 6:** Ethical concerns.
>
> Thank you for raising these important ethical concerns.
> To address data privacy, federated learning or cross-domain modeling can be integrated to enhance Coral's privacy-preserving capabilities. For fairness, we will ensure equitable outputs by identifying and mitigating the impact of sensitive attributes on model predictions. Additionally, LLM-based agents are expected to maintain learner autonomy and agency. We will include a discussion of these ethical considerations in the revised version of the paper.
>
> > **Question 1:** Dataset selection criteria.
>
> The data selection criterias include:
> 1. Authenticity: We use 3 real-world, open-source datasets, as detailed in Appendix D. These datasets reflect real learner interactions.
> 2. Diversity: Our datasets span different educational domains. The ASSISTments dataset targets K-12 education, while the Junyi Academy and NIPS datasets focus on online learning environments with global participants. This ensures coverage of varied learning contexts.
> 3. Representativeness: These datasets are established benchmarks in CD, widely used in current research for consistent evaluation.
>
> Obviously, our datasets adequately represent diversity of learning environments and learner behaviors.
>
> > **Question 2:** Baseline descriptions.
>
> We selected baselines from three aspects:
>
> 1. Educational Psychology Methods: IRT and MIRT, foundational in cognitive diagnosis,diagnose learners' cognitive states using psychology priors.
>
> 2. Machine Learning Approaches: PMF, NCDM, KaNCD, and RCD represent key methods using machine learning techniques to model student-item interactions.
>
> 3. Variational or Disentangled Methods: DCD and ReliCD, which are related to our solution.
>
> We will add detailed descriptions of each baseline in the Appendix of the revised version.
>
> ---
>
> [1] Disentangling cognitive diagnosis with limited exercise labels. NeurIPS'24.
>
> [2] Hiercdf: A bayesian network-based hierarchical cognitive diagnosis framework. SIGKDD'22.
>
> [3] RCD: Relation map driven cognitive diagnosis for intelligent education systems. SIGIR'21.

---

> > ### Comment · Reviewer_vXdf · 2024-08-13
> >
> > Thank you for your response. The additional steps 1,2 and 3 in the public response provided by the authors definitely increase the paper's overall quality a bit, which I am reflecting in my slight upgrade in ratings. Thanks.

---

> > > ### Author Response · Authors · 2024-08-13
> > >
> > > Thank you for your valuable comments and for recognizing the value of our work!
> > >
> > > We are pleased to have addressed your concerns and will revise the paper according to your suggestions. If you have any other questions, please feel free to ask. We will do our best to address any concerns you may have.
> > >
> > > Once again, we sincerely appreciate your encouragement and support.

---

### Author Rebuttal · Authors · 2024-08-05

**Public Response to All Reviewers**
---

---

We would like to express our thanks to the reviewers for their thorough reading of the paper and insightful comments. We first add some common experiments as suggested by the reviewers.

> **1. Additional baselines**

We conduct experiments on the four suggested baselines: VIBO[1], AGCDM[2], VarFA[3], and SCD[4], using three datasets. Each model is implemented based on their open-source code with minor modifications. For VIBO[1] and VarFA[3], we use 2PL (multi-dimensional) IRT as the backbone. AGCDM[2] requires modifications to handle our datasets, as its original code cannot distinguish between incorrect answers (negative samples) and unanswered questions. We mark unattempted records as -1, correct answers as 1, and incorrect answers as 0, and filter out unanswered data by adding the line of code “*data = np.array([entry for entry in data if entry['score'] != -1])*” in line 63 of AGCDM code's run.py file. For SCD[4], we use an open-source tool to construct the concept map and set feature vector dimensions for students and questions to match Coral. The results are as follows:

|ASSIST|ACC|AUC|RMSE|junyi|ACC|AUC|RMSE|NIPS2020EC|ACC|AUC|RMSE|
|-|-|-|-|-|-|-|-|-|-|-|-|
|VIBO|0.70265|0.70229|0.45342|VIBO|0.81265|0.79592|0.36928|VIBO|0.69667|0.73669|0.45723|
|AGCDM|0.70935|0.67705|0.45120|AGCDM|0.79310|0.66405|0.40817|AGCDM|0.70266|0.70318|0.45332|
|VarFA|0.71034|0.74630|0.44312|VarFA|0.81708|0.80347|0.37102|VarFA|0.71221|0.75224|0.44774|
|SCD|0.72003|0.75630|0.43102|SCD|0.81172|0.82037|0.37015|SCD|0.71013|0.78556|0.43617|
|**Coral**|**0.72533**|**0.77312**|**0.42034**|**Coral**|**0.82534**|**0.83503**|**0.36403**|**Coral**|**0.71622**|**0.79103**|**0.43200**|

The results demonstrate that Coral consistently outperforms the baselines across all datasets. Among the baselines, AGCDM's performance is impacted by the imbalanced distribution of positive and negative samples in the ASSIST and Junyi datasets (Please refer to Table 3 of our paper for specific statistics), resulting in high ACC but low AUC. In contrast, VIBO, VarFA, and SCD mitigate the impacts of imbalanced data through their use of variational inference and graph modeling techniques.

> **2. Hyper-parameter sensitivity experiments on embedding  dimensions**

We conduct additional hyper-parameter sensitivity experiments on different embedding dimensions $d$, with values of 124, 200, 256, 521, and 1024. Coral's performance on ACC is shown below:

|Value of $d$|124|200|256|512|1024|
|-|-|-|-|-|-|
|ASSIST|0.72667|0.72533|0.72411|0.71882|0.70611|
|junyi|0.82344|0.82134|0.81920|0.81474|0.82256|
|NIPS2020EC|0.72083|0.71622|0.72072|0.71334|0.71227|

We find that within an appropriate range, the dimension $d$ does not significantly affect the Coral's performance.

> **3. Add a new dataset (Optional suggestion)**

Before introducing a new dataset, we would like to first emphasize that our existing datasets are diverse, covering both K-12 scenarios (ASSIST) and online learning scenarios (Junyi and NIPS2020EC). These datasets are also representative, as they are benchmark datasets in cognitive diagnosis (CD) research. Nearly all top-tier work (e.g., [4][6][7]) uses these datasets for experiments.

To further demonstrate Coral's performance in different scenarios, we supplement our experiments with the PISA 2015[5] dataset. This dataset includes students' practice data from various countries and regions. We extract records of 1,000 learners from the Asian region, focusing on 200 questions. We filter out students with fewer than 20 response records, resulting in 21,814 practice records. The performance of Coral and representative baselines on this new dataset is shown below:

|Model|ACC|AUC|RMSE|
|-|-|-|-|
|IRT|0.67527|0.73416|0.46031|
|MIRT|0.62210|0.66973|0.48431|
|NCDM|0.65822|0.71403|0.49013|
|KANCD|0.67373|0.74271|0.45364|
|RCD|0.67234|0.72673|0.46253|
|**Our Coral**|**0.68312**|**0.74807**|**0.45217**|

The experimental results demonstrate that Coral maintains its performance advantage.

---

[1] https://arxiv.org/pdf/2002.00276

[2] https://ieeexplore.ieee.org/document/10027634

[3] https://arxiv.org/pdf/2005.13107

[4] Self-supervised graph learning for long-tailed cognitive diagnosis. AAAI'23.

[5] https://github.com/bigdata-ustc/EduData

[6] Disentangling cognitive diagnosis with limited exercise labels. NeurIPS'24.

[7] RCD: Relation map driven cognitive diagnosis for intelligent education systems. SIGIR'21.

---

### Decision · Program_Chairs · 2024-09-25

**Decision:**

Accept (poster)

**Comment:**

This paper introduces Coral, a Collaborative Cognitive Diagnosis model utilizing Disentangled Representation Learning. It enhances cognitive diagnosis by dynamically capturing collaborative signals among learners, leading to improved performance over state-of-the-art methods in several educational datasets. The proposed methodology shows promise in advancing personalized education through more nuanced insights into learners’ cognitive states.

**Strengths**
- Novelty: The paper introduces a unique approach by integrating collaborative signals among learners using disentangled representation learning. This allows for a better understanding of learners’ cognitive states and could enhance personalized educational interventions (Reviewers vXdf, bUz3).
- Strong Empirical Results: The model demonstrates superior performance across various real-world datasets, outperforming state-of-the-art methods in several metrics, including ACC, AUC, F1-score, and RMSE, highlighting its practical utility in both cold-start and sparse data scenarios (Reviewers Ty5J, t9qg).
- Explainability: By disentangling cognitive states, the Coral model provides greater explainability and controllability in diagnosing students’ cognitive states, which can be valuable for educators in understanding student learning patterns (Reviewers Ty5J, vXdf, UPTU).
- Collaborative Graph Learning: The model’s iterative graph construction process allows it to dynamically search for optimal neighbors, enhancing the model’s capability to leverage collaborative signals (Reviewers Ty5J, vXdf).

**Areas for Improvement (Weaknesses)**
- Computational Inefficiency: The model suffers from high computational costs, which may hinder its scalability and practical application in large-scale educational settings (Reviewers Ty5J, UPTU, bUz3).
- Limited Generalizability: The choice of datasets may limit the model’s generalizability to other educational contexts, as the current experiments focus on specific scenarios (Reviewers bUz3, Ty5J, t9qg).
- Weak Interpretability Evaluation: Although the model claims improved explainability, the visualizations (such as t-SNE) may not be sufficient to provide meaningful interpretability. More rigorous qualitative analysis and ablation studies are needed to evaluate the disentanglement quality (Reviewers Ty5J, bUz3).
- Sensitivity to Hyperparameters: The model’s performance is highly sensitive to hyperparameters like the number of neighbors (K) and beta value. This reliance could make it difficult for practitioners to fine-tune the model for different datasets (Reviewer Ty5J).

The Authors' actively and extensively participated in the rebuttal and Author-Reviewer discussion phase. They acknolwedged and addressed most Reviewer and AC concerns and adhered to most of the reasonable and valuable suggestions provided by the Reviewers and AC. This strengthens their work significantly from its initial state.

An overview of key concerns categorized by the degree to which they were addressed is listed below:

.1. Addressed Concerns:
- Data diversity and additional baselines: The Authors added a new dataset and ran additional experiments on suggested baselines like VIBO, AGCDM, VarFA, and SCD, with Coral outperforming them consistently. This strengthened the original claims about Coral’s superior performance.
- Hyper-parameter sensitivity: They conducted additional sensitivity experiments for different embedding dimensions, which showed Coral’s performance was stable within a reasonable range.
- Interpretability: The Authors clarified that disentanglement improved interpretability by aligning cognitive states and collaborative states and offered clearer explanations of how Coral’s predictions could be interpreted by educators.
2. Partially Addressed Concerns:
- Overfitting: While the Authors explained that they used early stopping and conducted experiments on data sparsity and cold-start scenarios, concerns about overfitting were acknowledged but not fully explored with extensive validation metrics.
- Explainability: The Authors provided examples of explainability in Appendix E, and additional conclusions were offered to support their claim. However, Reviewer concerns about limited exploration of how individual cognitive states relate to knowledge concepts were only partially addressed.
3. Negatively Addressed or Unaddressed Concerns:
- Computational efficiency: The Authors admitted the model’s inefficiency but offered practical solutions without significant exploration of how these might affect large-scale applications. This limitation may affect the practical deployment of Coral in real-world, large-scale settings.
- Ethical considerations: The Author response included a discussion of potential privacy and fairness concerns but lacked a detailed ethical framework or proposed safeguards for applying the model in real-world education systems.

Overall, the Author rebuttals generally improved the submission by addressing key concerns raised in the reviews, although some areas, particularly around computational efficiency and ethical considerations, remained only partially addressed.

Key considerations for my recommendation include (1-3 are positive, 4 is a remaining weakness, but has been acknowledged by the Authors):

1. Author Rebuttal Quality: The Authors have thoroughly addressed most of the Reviewers’ concerns. For instance, they provided new baseline comparisons, further hyper-parameter sensitivity analyses, and new datasets, including the PISA 2015 dataset. These additions strengthened the evidence for the model’s robustness, handling of cold-start scenarios, and disentanglement quality.
2. Reviewer Feedback: The Reviewers generally upgraded their ratings after the Author rebuttal phase, reflecting a consensus toward acceptance. Reviewer Ty5J increased their score after the rebuttal, acknowledging improvements in the paper.
3. Strengths Reinforced: The novelty and robustness of the Collaborative Cognitive Diagnosis model with Disentangled Representation Learning (Coral) was strongly supported by the empirical evidence provided by the Authors. The Authors successfully demonstrated the model’s generalizability, performance on diverse datasets, and its advantage over existing state-of-the-art methods.
4. Remaining Weaknesses: While computational inefficiencies and concerns about scalability remain, the Authors have proactively addressed these limitations with proposed solutions, and these do not appear to detract significantly from the overall contribution of the work.

The Author rebuttal phase was constructive, with several key concerns being effectively addressed. However, to further strengthen the submission, the Authors should address the identified weaknesses, particularly in terms of computational efficiency, ethical considerations, and further hyperparameter sensitivity analyses.

The paper makes positive contributions to collaborative* cognitive diagnosis with its novel approach, technical rigor, and strong empirical results. Overall, Coral’s impact on the field of educational data mining and cognitive diagnosis is promising. Regardless of whether the paper is accepted, it is strongly suggested that the Authors implement the changes that they agreed to for their paper. If the paper is accepted, then the Authors are expected to do so.